# Mineralogical, Textural and Chemical Characteristics of Ophiolitic Chromitite and Platinum Group Minerals from Kabaena Island (Indonesia): Their Petrogenetic Nature and Geodynamic Setting

**Arifudin Idrus [1],[*], Sara Septiana [2], Federica Zaccarini [3], Giorgio Garuti [3] and Hasria Hasria [2]**

[1] Department of Geological Engineering, Universitas Gadjah Mada, Jl. Grafika 2 Bulaksumur, Yogyakarta 55281, Indonesia

[2] Study Program of Geological Engineering, Universitas Halu Oleo, Kampus Hijau Bumi Tridharma, Anduonohu, Kendari 93231, Indonesia; saraseptiana10@gmail.com (S.S.); hasriageologi@gmail.com (H.H.)

[3] Department of Applied Geosciences and Geophysics, University of Leoben, Peter Tunner Str. 5, 8700 Leoben, Austria; federicazaccarinigaruti@gmail.com (F.Z.); giorgio.garuti1945@gmail.com (G.G.)

[*] Correspondence: arifidrus@ugm.ac.id; Tel.: +62-274-513668

**Abstract:** This contribution presents the first systematic mineralogical study of chromite composition, silicates and PGM (platinum group minerals) by electron microprobes in the podiform chromitite of Kabaena Island (Indonesia) mined in the past. The main target of this study is to understand the petrogenetic nature of the parental melt from which the chromitites of Kabaena Island precipitated and, indirectly, define the geodynamic tectonic setting of their emplacement. The evolution of PGM, from the magmatic stage to low-temperature processes, is also discussed. The variation of the $Cr\# = Cr/(Cr + Al)$, being comprised between 0.65 and 0.75, is similar to the podiform-type chromitite and indicates the absence of Al-rich chromite. The calculated composition of the parental melt varies from arc to MORB (mid-ocean ridge basalts). Several grains of olivine and clinopyroxene have been found in the silicate matrix or included in fresh chromite. Olivine shows a composition typical of a hosted mantle, and clinopyroxene is similar to those analyzed in the forearc of an SSZ (supra-subduction zone). Small PGM, varying in size from 1 to 10 μm, occur in the chromitites. The most abundant PGM is laurite, which has been found included in fresh chromite or in contact with ferrian chromite along the cracks in the chromite. Laurite forms polygonal crystals, and it occurs as a single phase or in association with clinopyroxene and amphibole. Tiny blebs of Ir-Os alloy (less than 2 μm across) have been found associated with grains of awaruite in the serpentine gangue of the chromitites. The composition of the investigated chromitites suggests that they formed in the mantle of a forearc ophiolite. All the discovered grains of laurite are considered to be magmatic in origin, i.e., entrapped as solid phases during the crystallization of chromite at temperatures above 1000 °C and a sulfur fugacity below sulfur saturation. Iridium–osmium alloys are secondary in origin and represent a low-temperature, around 400 °C, exsolution product.

**Keywords:** chromitite; olivine; clinopyroxene; platinum group minerals; ophiolite; Kabaena Island; Indonesiaz

## 1. Introduction

Most of Indonesian ophiolites are tectonically dismembered and related to the Eurasian, Indo-Australian and Pacific triple-active junction throughout the Late Mesozoic and Early Tertiary [1,2]. Therefore, due to the fact that they occur in a complex active geodynamic system, the Indonesian ophiolites formed and were emplacement in different tectonic settings. Some of them, cropping out in Java, Sumatra and Borneo, represent the Tethyan Ocean and were generated in a supra-subduction zone (SSZ), before the Late Cretaceous [1,2]. Those of Sulawesi, Halmahera and Papua likely belong to the so-called Circum Pacific

Phanerozoic polygenetic ophiolite belt [3–6]. These ophiolites, being tectonically dismembered and affected by alterations, host oceanic materials, different in origin, including fragments of rocks representing seamounts, oceanic plateaus, forearc and the mid-ocean ridge (MOR) [1–7].

The East Sulawesi ophiolite, covering an area of more than 10,000 km$^2$, is one of the three largest ophiolites in the world [8]. It occurs in the northeastern and southeastern arms of Sulawesi, and although tectonically disrupted, the East Sulawesi ophiolite constitutes an almost complete ophiolite suite. It consists, from base to top, of: (1) residual mantle peridotite, mostly spinel lherzolite intercalated with minor harzburgite and dunite; (2) mafic-ultramafic cumulate, including layered and isotropic gabbro; and (3) sheeted dolerites and basaltic volcanic rocks [8]. Hamilton [9] and Kadarusman et al. [10] described the following four main geological zones in Sulawesi (Figure 1): (1) the West and North Sulawesi Volcano-Plutonic Arc, Cenozoic in age, that comprises volcanic and plutonic rocks and an older Mesozoic basement complex composed of metamorphic and ultramafic rocks; Tertiary and Quaternary sediments have been also recognized; (2) the Central Sulawesi metamorphic belt made up of high-pressure metamorphic rocks, the Pompangeo schists, and an ophiolite mélange [11]; (3) the East Sulawesi ophiolite complex; (4) and the continental basement of Banggai Sula and Tukang Besi.

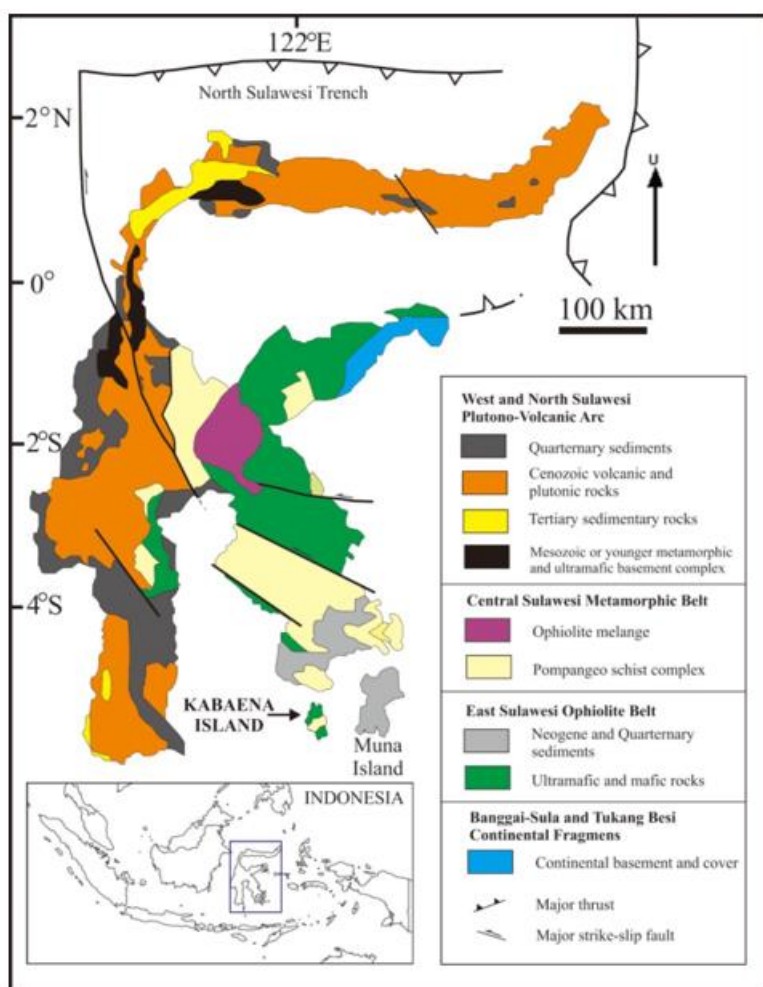

**Figure 1.** Geological sketch map of Sulawesi showing the location of Kabaena Island (simplified after Hamilton [9] and Kadarusman et al. [10]).

The genesis of the East Sulawesi ophiolite is still controversial. According to Soeria-Atmadja et al. [12] and Simandjuntak [13], it represents a typical MOR (mid-ocean ridge) ophiolite, and later, Monnier et al. [8], Bergman et al. [14], and Parkinson [15] suggested a

SSZ setting. More recently, Kadarusman et al. [10] observed great differences in the rock composition and mineral chemistry of several peridotites collected in different outcrops of East Sulawesi ophiolite. Furthermore, the REE (rare earth elements) distribution in clinopyroxene from peridotite, the crystallization sequence of gabbroic rocks, and major and trace element geochemistry of basalt and dolerite are similar to those formed in a MOR setting [10]. Therefore, Kadarusman et al. [10] proposed a more complex model for the formation of East Sulawesi ophiolite, including MOR, oceanic plateau, and to a lesser extent, SSZ geodynamic settings.

For many decades, chromite composition has been used to distinguish the two main types of chromitite, i.e., podiform associated with a residual mantle in ophiolite belts and stratiform in continental layered intrusions [16–24]. More rarely, small stratiform chromitite formed in the cumulates above the Moho in the ophiolite [25].

It has been demonstrated that the composition of chromite and associated silicates from podiform chromitites is related to the type of parental magma, such as mid-ocean ridge basalt (MORB), oceanic island basalt (OIB), and boninite, as well as to the degree of partial melting of the mantle source. Therefore, chromitite has become a reliable indicator of the geodynamic setting in which the minerals and their host rock have formed [16–25].

Podiform chromitites very often host tiny grains, generally less than 10 μm in size, of platinum group minerals (PGM). The PGM are the natural carriers of the six noble platinum group elements (PGE) that comprise Os, Ir, Ru, Rh, Pt, and Pd. The PGM associated with podiform chromitites generally consists of Os-Ir-Ru phases, and they occur included in chromite crystals or in the interstitial silicate of the chromitite. The PGM enclosed in unaltered chromite are preserved by alteration, and they crystallized at high temperatures in the mantle during the magmatic stage [26–28]. On the contrary, the PGM not included in chromite are much more exposed to alteration and can be reworked and remobilized on a small scale by the action of low-temperatures hydrous fluids [28–31]. Despite being numbered as one of the three largest ophiolites in the world [8], the East Sulawesi ophiolite hosts a limited number of documented chromitite occurrences because of poor accessibility and the presence of impenetrable tropical rain forest.

The Island of Kabaena, located in Southeast Sulawesi Province, is characterized by a complex tectonic evolution caused by the collision between the microcontinent and oceanic plates. The oceanic plate contains fragments of mantle peridotites that belong to the East Sulawesi ophiolite. The mantle peridotites of Kabaena Island host small bodies of podiform chromitite that have been only recently and partially studied [32].

In this contribution, we report a first systematic study of chromite, silicates, and PGM composition by electron microprobe in the podiform chromitite of Kabaena Island. Since the origin of Kabaena ultramafic rocks is still a matter of discussion [10], and due to the complicated outcropping conditions that limited the field observation, the data are used to better understand the petrogenetic nature of the parental melt from which the chromitites of Kabaena Island precipitated and, indirectly, define the geodynamic tectonic setting of their emplacement. The evolution of PGM, from the magmatic stage to low-temperature processes, is also discussed.

## 2. Geology of Kabaena Island and Description of the Studied Chromitites

### 2.1. Geology

Kabaena Island, covering an area of about 900 km$^2$, is a satellite island of mainland Sulawesi, located in the southern part of the Southeast Sulawesi Arm (Figure 1). Although the geochronological data available for Kabaena Island are limited, on the basis of the reconstruction provided by Hall and Sevastjanova [33], the age of Kabaena Island varies from the Cretaceous to the Middle Miocene ($\pm$15 Ma). Kabaena Island, along with Muna Island, are separated by a subduction zone from the Banda Peninsula. Kabaena Island was also pushed closer to the Southeast Sulawesi Arm due to the division of the South Banda Sea during the Pleistocene ($\pm$5 Ma). According to Simandjuntak [34], Kabaena

Island is part of the East Ophiolite Sulawesi Belt and mostly consists of ultramafic rocks, representing a residual mantle, Cretaceous in age (Figure 2A,B).

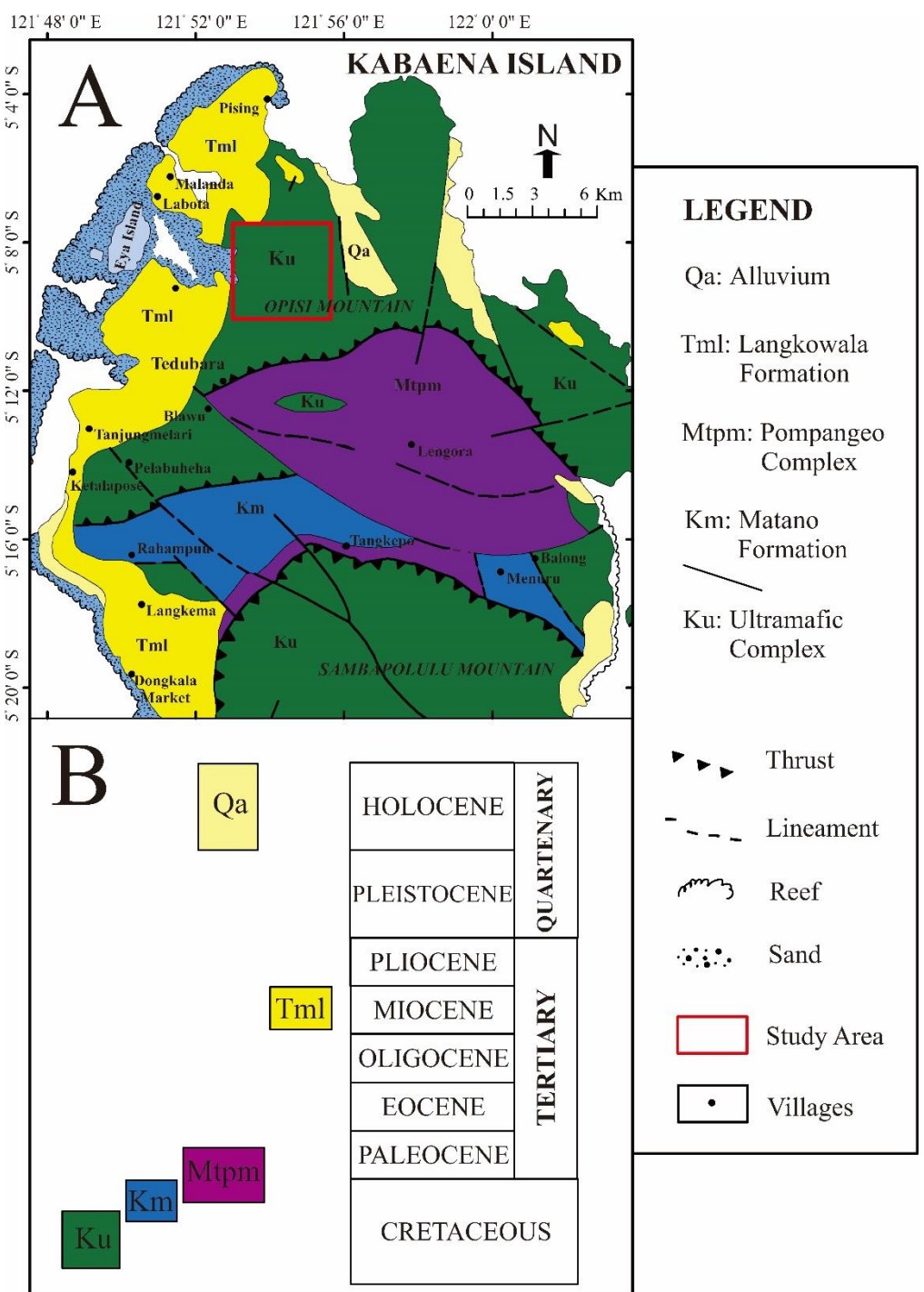

**Figure 2.** Geological map of the northern part of Kabaena Island showing the study area (**A**) and stratigraphy (**B**), (modified after Simandjuntak [34]). Light blue with dots area is inferred coral reef occurrence. See the text for the detailed description of the formation and complex.

In the northern area of Kabaena Island, the ultramafic rocks are exposed in the Opisi and Sambapolulu Mountains and occur in contact with the Pompangeo Schist Complex and the Matano and Langkowala Formations (Figure 2A,B). The Pompangeo Schist Complex underlies the peridotite and contains low-moderate grade metamorphic rocks comprising phyllite, mica schist, amphibole schist, and chlorite schist and is Cretaceous and Tertiary

in age (Figure 2B). The Matano formation consists of Cretaceous limestone (Figure 2B) containing calcite and dolomite crystallized during a diagenetic process. The limestone has been locally metamorphosed, forming meta-limestone and marble. The Matano Formation also contains minor black shale that is generally transformed into slate. The Langkowala Formation, Tertiary in age (Figure 2B), consists of a conglomerate containing ultramafic and metamorphic rocks, sandstones, and minor calcarenite [35]. Quaternary alluvium sediments are widespread along the coast of Kabaena Island (Figure 2A,B). The ultramafic rocks of East Sulawesi ophiolite are exposed in the northern and southern parts of Kabaena Island [32,34]. Based on local geological mapping, most of the peridotites are fine to coarse-grained lherzolite, accompanied by minor harzburgite, which are partially unconformably covered by Tertiary sediments such as limestone [32] (Figure 3).

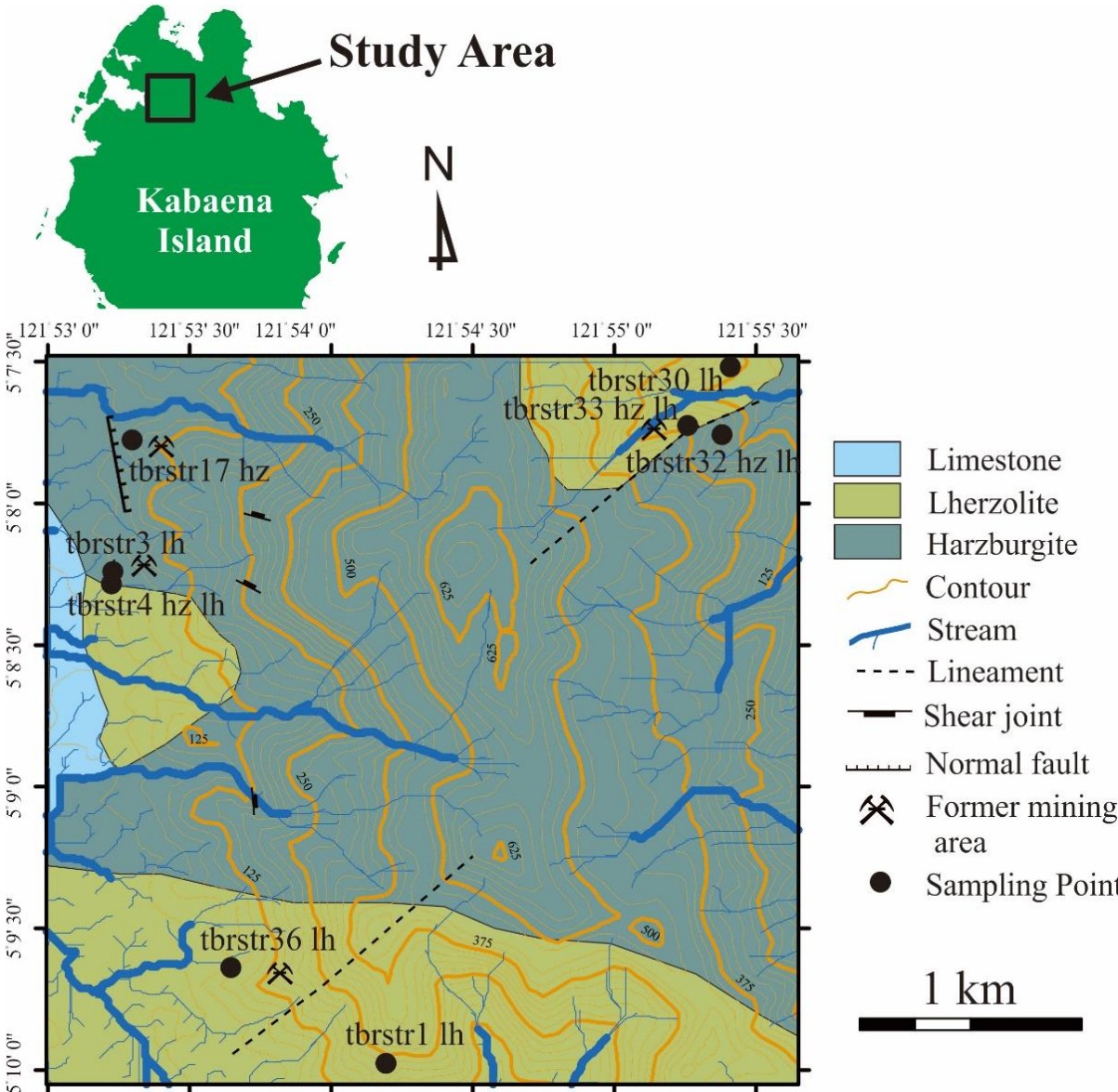

**Figure 3.** Detailed geological map of the study area (modified after Septiana et al. [32]) and sample location. Geological map of the northern part of Kabaena Island showing the study area (modified after Simandjuntak [34]).

## 2.2. Material

The studied chromitites were collected in eight areas from the northern part of Kabaena Island in the Tedubara area (Figures 2A and 3). They occur in remote areas with difficult

access because of the dense vegetation and high hilly morphology [30]. They were mined for chromite by the local community in the past, but now, all the mining activities are ceased. Chromitites are hosted in an altered lherzolite and harzburgite (Figure 4A,B) and form small lens and schlieren bodies (Figure 4C–E), and in some cases, they show the nodule texture (Figure 4F). Four samples (tbrstr3 hz lh, tbrstr4 hz lh, tbrstr 32 hz lh, and tbrstr33 hz lh) were collected in the contact between harzburgite and lherzolite; three chromitites (tbrstr1 lh, tbrstr30 lh and tbrstr36 lh) were sampled in the lherzolite; and one sample (tbrstr17 hz) in the harzburgite (Figure 3).

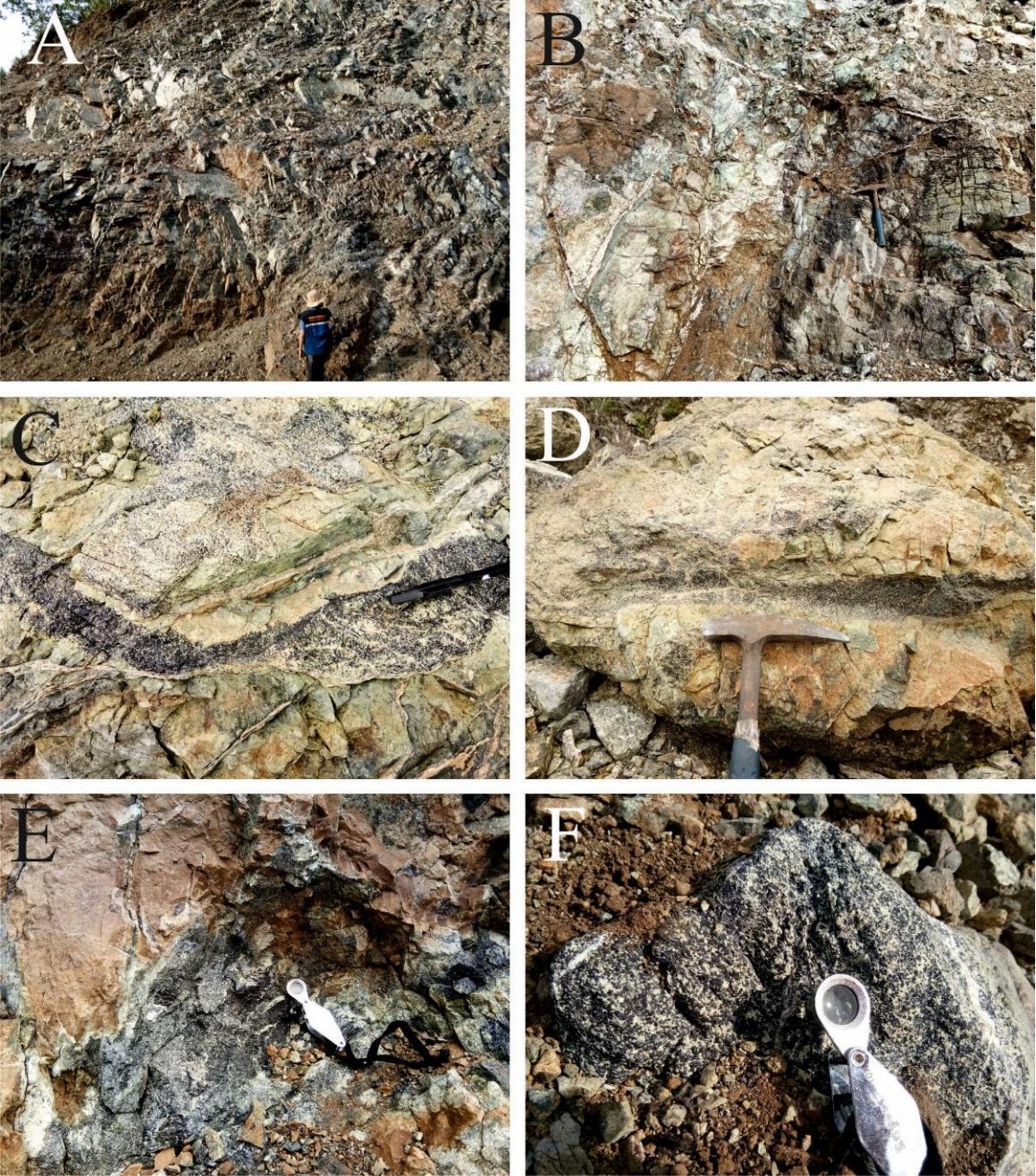

**Figure 4.** Field images of the Kabaena Island chromite deposits studied in this work. Overview of the altered peridotite hosting small bodies of chromitite (**A**,**B**), Schlieren-type and vein-shaped chromitite (**C**,**D**), a massive pod of chromitite (**E**), and nodular chromitite (**F**).

## 3. Methods

Two polished sections were prepared from each sample of cromitite and examined by reflected-light microscopy in order to study the chromite texture and to find the PGM. After, chromite, silicates, PGM, and other accessory minerals were analyzed by electron microprobe using a Superprobe JEOL JXA 8200 installed in the Eugen F. Stumpfl Laboratory at the University of Leoben, Austria using both ED and WD systems. Back-scattered electron (BSE) images were obtained using the same instrument. During the quantitative analyses of chromite and silicates, the electron microprobe was operated in the WDS mode, with an accelerating voltage of 15 kV of and beam current of 10 nA. The analysis of Na, Mg, K, Al, Si, Ca, Ti, V, Cr, Zn, Mn, Fe, Co, and Ni were obtained using the K$\alpha$ lines and were calibrated on natural chromite, rhodonite, ilmenite, albite, pentlandite, wollastonite, kaersutite, sphalerite, skutterudite, and metallic vanadium. The following diffracting crystals were used: TAP for Na, Mg, and Al; PETJ for K, Si, and Ca; and LIFH for Ti, V, Cr, Zn, Mn, Fe, Co, and Ni. The peak and background counting times were 20 and 10 s, respectively, for the major elements. They were increased to 40 and 20 for the trace elements, such Ca, Ni, and Mn in olivine. The detection limits were automatically calculated by the microprobe software, and they are listed in the following as ppm: Ca (50); K (80); Mg, Mn, and Fe (100); Na, V, Ni, and Co (150); Cr and Zn (200); and Al, Si, and Ti (250). The amount of $Fe^{3+}$ in chromite was calculated assuming the ideal spinel stoichiometry. The accessory minerals smaller than 10 µm were only qualitatively analyzed by EDS. The larger PGM were quantitatively analyzed in the WDS mode, at 20 kV accelerating voltage and 10 nA beam current, and a beam diameter of about 1 micron. The peak and background counting times were 15 and 5 s, respectively. The K$\alpha$ lines were used for S and As; L$\alpha$ for Ir, Ru, Rh, and Pd; and M$\alpha$ for Os. The reference materials were pure metals for the PGE (Ru, Rh, Pd, Os, and Ir); natural pyrite for S; and platarsite for As. The following diffracting crystals were selected: PETJ for S; PETH for Ru, Os, and Rh; LIFH for Ir; and TAP for As. Automatic corrections were performed for interferences involving Ru–Rh and Rh–Pd.

## 4. Results

### 4.1. Chromitite Texture and Chromite Composition

Microscopically, the investigated samples can be classified as massive chromitite, containing from 60 to 90 volume % of chromite (Figure 5A–D). They display the following textures: massive and fractured and lobate (Figure 5A–D). Chromite is usually fresh, showing a rarely narrow magnetite alteration rim only along the grain boundaries and cracks (Figure 5C). The silicates of the gangue mainly consist of serpentine, talc, and minor chlorite. Preserved crystals of olivine and clinopyroxene have also been found (Figure 5D). Four hundred electron microprobe analyses were performed on 16 samples from the eight chromitite occurrences, and a selection of them is listed in Table 1. Chromites are compositionally homogeneous or display little variations within single localities. However, significant differences are observed from one locality to the other, especially in the chromitite occurring in the contact between the harzburgite and lherzolite. The average (av) variation ranges and the minimum (min) and maximum (max) values of the major oxides (wt%), Cr# = (Cr/Cr + Al), and Fe# = ($Fe^{2+}/Fe^{2+}$ + Mg) in chromite grains are listed in the following: $Cr_2O_3$ = av 48.42, min 47.46, max 49.63; $Al_2O_3$ = av 20.74, min 19.85, max 21.34; MgO = av 13.38, min 12.88, max 13.84; FeO = av 13.27, min 14.07, max 14.43, Cr# = av 0.61, min 0.6, max 0.62; and Fe# = av 0.39, min 0.37, max 0.41; in the chromitite from harzburgite; $Cr_2O_3$ = av 50.66, min 46.24, max 56.35; $Al_2O_3$ = 18.08, min 12.41, max 21.11; MgO = av 14.59, min 10.37, max 15.21; FeO = av 13.31, min 12.81, max 20.91; Cr# = av 0.65, min 0.65, max 0.75, and Fe# = av 0.38, min 0.33, max 0.53; in the chromitite from lherzolite; $Cr_2O_3$ = av 53.14, min 48.62, max 56.32; $Al_2O_3$ = 16.49, min 14.08, max 19.86; MgO = av 13.46, min 11.99, max 14.76; FeO = av 14.17, min 12.5, max 16.58; Cr# = av 0.68, min 0.62, max 0.72, and Fe# = av 0.39, min 0.32, max 0.44; and in the chromitite from the contact, harzburgite-lherzolite. The calculated amount of $Fe_2O_3$ is comprised between 0 and 4 wt%, and the values of $TiO_2$ are always below 0.36 wt%, independently of the host peridotite. The

SiO$_2$, ZnO, and NiO amounts are typically below the detection limit. Detectable contents of MnO (from 0.15 to 0.34 wt%) and of V$_2$O$_3$ (up to 0.2 wt%) have been determined. In the diagrams used to discriminate podiform from stratiform chromitites [16,36,37], the composition of Kabaena Island chromite plots if the field of the podiform was chromitites (Figure 6A–D). They display a limited compositional variation when compared with the chromitites of Sulawesi [38].

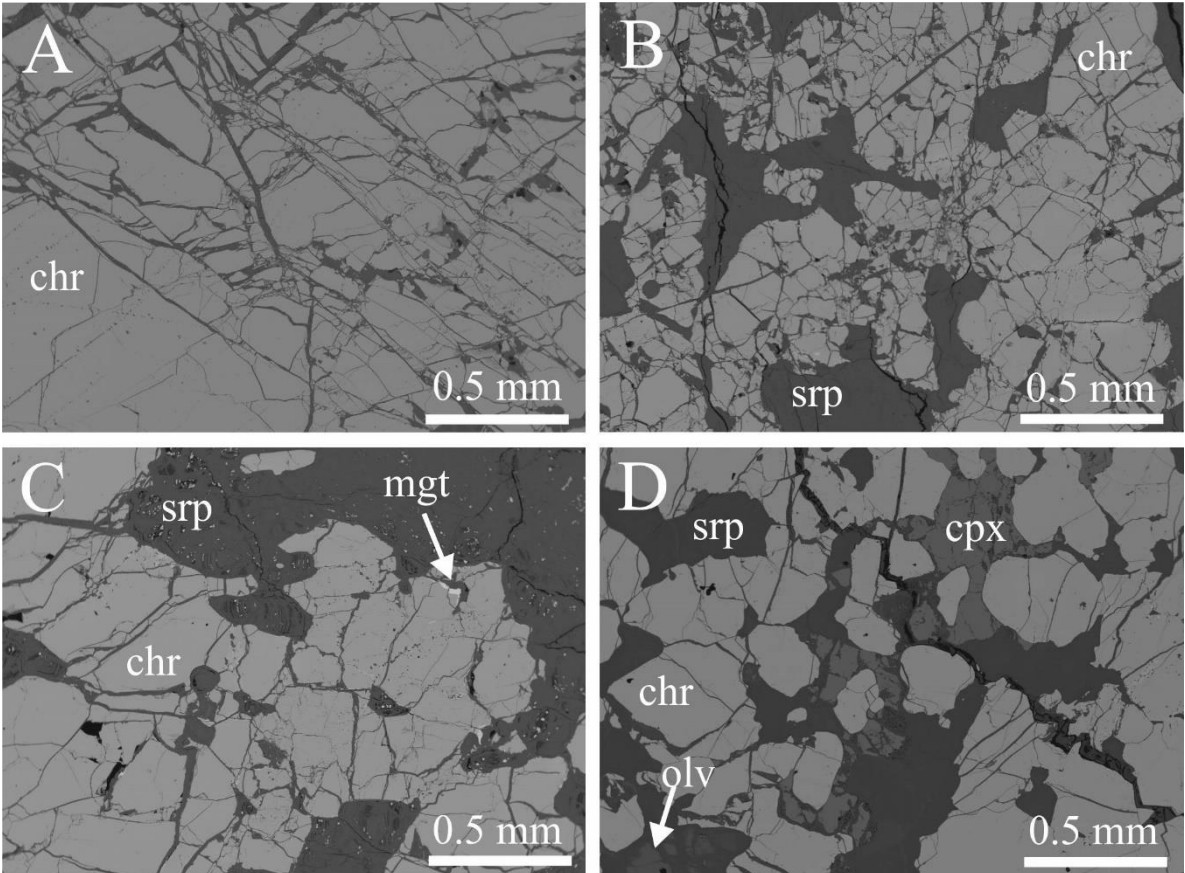

**Figure 5.** Back-scattered electron (BSE) images of the chromite texture. Massive chromitite composed of closely interlocked chromite crystals cut by cracks and fissures filled with serpentine (black) (**A**), fractured chromitite containing up to 30% interstitial silicate (**B**), lobate chromitite showing small spots of magnetite alterations (**C**), and preserved clinopyroxene and olivine occurring in the chromitite matrix (**D**). Abbreviations: chr = chromite, srp = serpentine, mgt = magnetite, cpx = clinopyroxene, and olv = olivine.

However, in the binary diagram Cr# versus Fe# (Figure 6A), some analyses fell in the overlapping fields of the stratiform, podiform chromite. Some samples showed a negative correlation between Cr$_2$O$_3$ and Al$_2$O$_3$ (Figure 6B) consistent with the Al-Cr substitution predominant in podiform chromitites. In other samples, consisting of chromitite hosted in lherzolite and in contact harzburgite-lherzolite, this correlation is not present (Figure 6D). All the samples are poor in TiO$_2$, as typical for the chromitite hosted in the mantle sequence of ophiolite complexes (Figure 6D).

In order to better understand the origin of Kabaena Island chromitites, the composition of Kabaena Island chromite has been plotted in the binary diagrams applied to distinguish the tectonic setting (Figure 7A–C) [18,23,37]. The TiO$_2$ versus Al$_2$O$_3$ contents of chromite indicate an affinity for the spinels from the SSZ peridotite (Figure 7A), as previously reported for the Sulawesi chromitite [38].

**Table 1.** Selected microprobe analyses of chromite (wt%) from chromitites of Kabaena Island ophiolite.

| Sample | SiO$_2$ | Cr$_2$O$_3$ | Al$_2$O$_3$ | TiO$_2$ | FeO | Fe$_2$O$_3$ | MgO | MnO | NiO | ZnO | V$_2$O$_3$ | Total | Fe# | Cr# |
|---|---|---|---|---|---|---|---|---|---|---|---|---|---|---|
| tbrstr1 lh | 0.02 | 48.41 | 20.31 | 0.32 | 13.23 | 3.34 | 14.50 | 0.21 | 0.08 | 0.00 | 0.08 | 100.51 | 0.34 | 0.62 |
| tbrstr1 lh | 0.00 | 48.53 | 20.32 | 0.32 | 13.27 | 2.87 | 14.36 | 0.25 | 0.04 | 0.01 | 0.11 | 100.09 | 0.34 | 0.62 |
| tbrstr1 lh | 2.03 | 46.24 | 19.34 | 0.28 | 14.18 | 1.56 | 15.21 | 0.20 | 0.08 | 0.00 | 0.06 | 99.18 | 0.34 | 0.62 |
| tbrstr1 lh | 0.02 | 48.66 | 20.17 | 0.29 | 13.54 | 2.72 | 14.13 | 0.22 | 0.10 | 0.00 | 0.09 | 99.93 | 0.35 | 0.62 |
| tbrstr1 lh | 0.03 | 48.97 | 20.24 | 0.30 | 14.14 | 2.27 | 13.79 | 0.24 | 0.13 | 0.00 | 0.14 | 100.25 | 0.37 | 0.62 |
| tbrstr1 lh | 0.00 | 49.16 | 20.31 | 0.21 | 13.89 | 2.55 | 13.93 | 0.25 | 0.09 | 0.02 | 0.03 | 100.42 | 0.36 | 0.62 |
| tbrstr1 lh | 0.00 | 48.82 | 20.16 | 0.28 | 13.48 | 2.67 | 14.17 | 0.21 | 0.08 | 0.01 | 0.10 | 99.99 | 0.35 | 0.62 |
| tbrstr1 lh | 0.04 | 49.02 | 20.22 | 0.29 | 13.83 | 2.88 | 14.17 | 0.20 | 0.11 | 0.00 | 0.09 | 100.86 | 0.35 | 0.62 |
| tbrstr1 lh | 0.01 | 49.23 | 20.28 | 0.36 | 13.47 | 2.76 | 14.44 | 0.24 | 0.06 | 0.01 | 0.11 | 100.96 | 0.34 | 0.62 |
| tbrstr3 hz lh | 0.06 | 54.93 | 14.75 | 0.18 | 13.90 | 2.62 | 13.32 | 0.24 | 0.08 | 0.02 | 0.00 | 100.10 | 0.37 | 0.71 |
| tbrstr3 hz lh | 0.00 | 55.27 | 14.40 | 0.24 | 13.91 | 2.50 | 13.26 | 0.26 | 0.04 | 0.01 | 0.09 | 99.97 | 0.37 | 0.72 |
| tbrstr3 hz lh | 0.01 | 55.17 | 14.36 | 0.22 | 14.26 | 2.30 | 12.98 | 0.17 | 0.12 | 0.00 | 0.10 | 99.68 | 0.38 | 0.72 |
| tbrstr3 hz lh | 0.00 | 55.91 | 14.55 | 0.20 | 14.82 | 2.07 | 12.80 | 0.28 | 0.05 | 0.04 | 0.06 | 100.76 | 0.39 | 0.72 |
| tbrstr3 hz lh | 0.03 | 55.53 | 14.44 | 0.20 | 14.17 | 2.07 | 13.08 | 0.19 | 0.13 | 0.00 | 0.03 | 99.89 | 0.38 | 0.72 |
| tbrstr3 hz lh | 0.00 | 55.78 | 14.49 | 0.24 | 14.54 | 2.15 | 12.99 | 0.26 | 0.06 | 0.00 | 0.06 | 100.58 | 0.39 | 0.72 |
| tbrstr3 hz lh | 0.10 | 55.78 | 14.49 | 0.25 | 14.16 | 2.35 | 13.38 | 0.25 | 0.07 | 0.00 | 0.05 | 100.87 | 0.37 | 0.72 |
| tbrstr3 hz lh | 0.00 | 54.85 | 14.24 | 0.25 | 14.55 | 2.16 | 12.65 | 0.22 | 0.10 | 0.00 | 0.04 | 99.06 | 0.39 | 0.72 |
| tbrstr3 hz lh | 0.01 | 56.10 | 14.51 | 0.22 | 13.95 | 2.19 | 13.36 | 0.28 | 0.11 | 0.00 | 0.02 | 100.75 | 0.37 | 0.72 |
| tbrstr3 hz lh | 0.08 | 55.50 | 14.34 | 0.27 | 14.38 | 1.89 | 12.98 | 0.27 | 0.08 | 0.01 | 0.05 | 99.85 | 0.38 | 0.72 |
| tbrstr4 hz lh | 0.00 | 55.63 | 14.70 | 0.17 | 12.94 | 2.70 | 13.98 | 0.23 | 0.11 | 0.01 | 0.10 | 100.58 | 0.34 | 0.72 |
| tbrstr4 hz lh | 0.00 | 56.63 | 14.94 | 0.23 | 13.45 | 2.36 | 14.03 | 0.23 | 0.11 | 0.01 | 0.07 | 102.06 | 0.35 | 0.72 |
| tbrstr4 hz lh | 0.02 | 56.15 | 14.51 | 0.23 | 12.88 | 2.31 | 14.07 | 0.24 | 0.08 | 0.02 | 0.05 | 100.55 | 0.34 | 0.72 |
| tbrstr4 hz lh | 0.02 | 55.97 | 14.47 | 0.27 | 15.44 | 0.00 | 14.16 | 0.21 | 0.09 | 0.00 | 0.09 | 100.71 | 0.38 | 0.72 |
| tbrstr4 hz lh | 0.09 | 55.69 | 14.36 | 0.25 | 13.65 | 1.72 | 13.42 | 0.24 | 0.10 | 0.00 | 0.08 | 99.60 | 0.36 | 0.72 |
| tbrstr4 hz lh | 0.08 | 55.87 | 14.40 | 0.28 | 13.14 | 2.21 | 13.91 | 0.31 | 0.04 | 0.01 | 0.12 | 100.37 | 0.35 | 0.72 |
| tbrstr4 hz lh | 0.04 | 55.89 | 14.40 | 0.21 | 13.57 | 1.87 | 13.50 | 0.23 | 0.09 | 0.01 | 0.14 | 99.94 | 0.36 | 0.72 |
| tbrstr4 hz lh | 0.03 | 56.48 | 14.55 | 0.20 | 13.42 | 2.09 | 13.80 | 0.21 | 0.10 | 0.00 | 0.05 | 100.92 | 0.35 | 0.72 |
| tbrstr4 hz lh | 0.04 | 56.37 | 14.49 | 0.27 | 15.25 | 0.00 | 13.60 | 0.21 | 0.06 | 0.00 | 0.05 | 100.34 | 0.39 | 0.72 |
| tbrstr4 hz lh | 0.03 | 56.37 | 14.45 | 0.27 | 15.12 | 0.00 | 13.94 | 0.24 | 0.12 | 0.01 | 0.08 | 100.64 | 0.38 | 0.72 |
| tbrstr17 lh | 0.00 | 48.86 | 20.11 | 0.25 | 13.89 | 2.62 | 13.85 | 0.30 | 0.07 | 0.01 | 0.12 | 100.07 | 0.36 | 0.62 |
| tbrstr17 hz | 0.00 | 47.58 | 21.27 | 0.16 | 14.58 | 2.66 | 13.50 | 0.25 | 0.11 | 0.00 | 0.09 | 100.22 | 0.38 | 0.60 |
| tbrstr17 hz | 0.00 | 48.00 | 21.34 | 0.21 | 14.98 | 2.16 | 13.42 | 0.22 | 0.00 | 0.02 | 0.13 | 100.50 | 0.39 | 0.60 |
| tbrstr17 hz | 0.00 | 48.03 | 21.13 | 0.17 | 15.28 | 2.18 | 13.13 | 0.23 | 0.00 | 0.00 | 0.10 | 100.24 | 0.40 | 0.60 |
| tbrstr17 hz | 0.01 | 48.54 | 20.57 | 0.14 | 15.00 | 1.72 | 13.00 | 0.28 | 0.10 | 0.00 | 0.15 | 99.50 | 0.39 | 0.61 |
| tbrstr17 hz | 0.03 | 48.76 | 20.65 | 0.17 | 14.60 | 2.05 | 13.50 | 0.25 | 0.05 | 0.01 | 0.11 | 100.18 | 0.38 | 0.61 |
| tbrstr17 hz | 0.10 | 48.21 | 20.42 | 0.21 | 15.21 | 2.00 | 13.03 | 0.25 | 0.04 | 0.01 | 0.06 | 99.54 | 0.40 | 0.61 |
| tbrstr17 hz | 0.12 | 48.32 | 20.21 | 0.17 | 14.69 | 2.21 | 13.29 | 0.23 | 0.09 | 0.01 | 0.13 | 99.46 | 0.38 | 0.62 |
| tbrstr17 hz | 0.08 | 48.93 | 20.46 | 0.17 | 15.42 | 2.08 | 13.06 | 0.26 | 0.05 | 0.01 | 0.08 | 100.59 | 0.40 | 0.62 |
| tbrstr17 hz | 0.00 | 49.36 | 20.57 | 0.13 | 15.22 | 2.08 | 13.23 | 0.21 | 0.02 | 0.00 | 0.10 | 100.93 | 0.39 | 0.62 |
| tbrstr30 lh | 0.03 | 54.73 | 13.39 | 0.20 | 16.67 | 3.35 | 11.38 | 0.27 | 0.05 | 0.02 | 0.13 | 100.23 | 0.45 | 0.73 |
| tbrstr30 lh | 0.04 | 55.59 | 13.59 | 0.20 | 15.53 | 2.14 | 12.01 | 0.27 | 0.09 | 0.01 | 0.08 | 99.56 | 0.42 | 0.73 |
| tbrstr30 lh | 0.03 | 55.29 | 13.51 | 0.23 | 15.42 | 2.97 | 12.22 | 0.28 | 0.06 | 0.01 | 0.13 | 100.15 | 0.41 | 0.73 |
| tbrstr30 lh | 0.12 | 55.56 | 13.56 | 0.24 | 15.34 | 2.46 | 12.38 | 0.24 | 0.08 | 0.00 | 0.20 | 100.18 | 0.41 | 0.73 |
| tbrstr30 lh | 0.06 | 55.68 | 13.36 | 0.17 | 16.24 | 3.15 | 11.78 | 0.31 | 0.09 | 0.01 | 0.08 | 100.93 | 0.44 | 0.74 |
| tbrstr30 lh | 0.00 | 55.09 | 13.22 | 0.16 | 16.86 | 3.71 | 11.31 | 0.25 | 0.06 | 0.04 | 0.13 | 100.83 | 0.46 | 0.74 |
| tbrstr30 lh | 0.05 | 54.10 | 12.91 | 0.21 | 16.88 | 3.94 | 11.06 | 0.30 | 0.08 | 0.01 | 0.10 | 99.63 | 0.46 | 0.74 |
| tbrstr30 lh | 0.02 | 55.43 | 13.10 | 0.22 | 17.06 | 3.30 | 11.28 | 0.23 | 0.01 | 0.03 | 0.17 | 100.84 | 0.46 | 0.74 |
| tbrstr30 lh | 0.00 | 54.94 | 12.63 | 0.17 | 20.19 | 0.00 | 10.67 | 0.30 | 0.03 | 0.01 | 0.14 | 99.10 | 0.52 | 0.74 |
| tbrstr30 lh | 0.07 | 54.76 | 12.41 | 0.18 | 20.91 | 0.00 | 10.37 | 0.30 | 0.09 | 0.01 | 0.17 | 99.27 | 0.53 | 0.75 |
| tbrstr32 hz lh | 0.04 | 49.61 | 19.23 | 0.25 | 15.63 | 2.81 | 12.79 | 0.34 | 0.08 | 0.02 | 0.12 | 100.91 | 0.41 | 0.63 |
| tbrstr32 hz lh | 0.02 | 50.25 | 18.62 | 0.27 | 15.96 | 1.89 | 12.35 | 0.28 | 0.05 | 0.00 | 0.13 | 99.82 | 0.42 | 0.64 |
| tbrstr32 hz lh | 0.00 | 49.68 | 18.39 | 0.23 | 15.38 | 2.61 | 12.54 | 0.26 | 0.08 | 0.00 | 0.08 | 99.25 | 0.41 | 0.64 |
| tbrstr32 hz lh | 0.04 | 50.43 | 18.66 | 0.23 | 16.15 | 2.36 | 12.43 | 0.23 | 0.09 | 0.00 | 0.13 | 100.76 | 0.42 | 0.64 |
| tbrstr32 hz lh | 0.02 | 50.56 | 18.71 | 0.24 | 16.40 | 2.18 | 12.28 | 0.21 | 0.07 | 0.01 | 0.09 | 100.77 | 0.43 | 0.64 |
| tbrstr32 hz lh | 0.03 | 50.32 | 18.58 | 0.27 | 16.44 | 1.78 | 12.11 | 0.23 | 0.04 | 0.00 | 0.13 | 99.95 | 0.43 | 0.64 |
| tbrstr32 hz lh | 0.04 | 50.29 | 18.53 | 0.25 | 15.83 | 2.36 | 12.50 | 0.24 | 0.12 | 0.01 | 0.10 | 100.25 | 0.42 | 0.65 |
| tbrstr32 hz lh | 0.04 | 50.35 | 18.46 | 0.23 | 16.26 | 1.89 | 12.16 | 0.21 | 0.09 | 0.00 | 0.12 | 99.80 | 0.43 | 0.65 |
| tbrstr32 hz lh | 0.00 | 50.93 | 18.67 | 0.22 | 16.45 | 1.84 | 12.19 | 0.25 | 0.07 | 0.01 | 0.10 | 100.73 | 0.43 | 0.65 |
| tbrstr32 hz lh | 0.00 | 50.31 | 18.37 | 0.22 | 15.68 | 2.33 | 12.45 | 0.27 | 0.10 | 0.00 | 0.15 | 99.87 | 0.41 | 0.65 |
| tbrstr33 hz lh | 0.03 | 51.33 | 18.25 | 0.26 | 12.63 | 3.50 | 14.76 | 0.24 | 0.09 | 0.00 | 0.09 | 101.19 | 0.32 | 0.65 |
| tbrstr33 hz lh | 0.02 | 50.94 | 18.10 | 0.30 | 12.66 | 3.33 | 14.55 | 0.27 | 0.08 | 0.00 | 0.09 | 100.33 | 0.33 | 0.65 |
| tbrstr33 hz lh | 0.06 | 51.17 | 18.17 | 0.20 | 12.50 | 2.88 | 14.55 | 0.23 | 0.17 | 0.00 | 0.09 | 100.03 | 0.33 | 0.65 |
| tbrstr33 hz lh | 0.13 | 51.30 | 17.69 | 0.20 | 13.08 | 3.00 | 14.23 | 0.22 | 0.11 | 0.00 | 0.12 | 100.07 | 0.34 | 0.66 |
| tbrstr33 hz lh | 0.03 | 52.02 | 17.92 | 0.28 | 13.59 | 2.19 | 13.93 | 0.20 | 0.19 | 0.00 | 0.06 | 100.42 | 0.35 | 0.66 |
| tbrstr33 hz lh | 0.00 | 52.28 | 17.80 | 0.22 | 13.57 | 2.16 | 13.88 | 0.22 | 0.14 | 0.00 | 0.10 | 100.37 | 0.35 | 0.66 |
| tbrstr33 hz lh | 0.04 | 52.26 | 17.76 | 0.21 | 13.56 | 2.09 | 13.86 | 0.27 | 0.13 | 0.00 | 0.08 | 100.25 | 0.35 | 0.66 |
| tbrstr33 hz lh | 0.06 | 51.63 | 17.49 | 0.23 | 13.78 | 2.27 | 13.64 | 0.15 | 0.08 | 0.02 | 0.09 | 99.45 | 0.36 | 0.66 |
| tbrstr33 hz lh | 0.02 | 52.29 | 17.71 | 0.24 | 13.62 | 2.07 | 13.87 | 0.23 | 0.05 | 0.01 | 0.07 | 100.19 | 0.36 | 0.66 |
| tbrstr33 hz lh | 0.05 | 51.61 | 17.35 | 0.26 | 13.32 | 2.81 | 13.98 | 0.20 | 0.08 | 0.01 | 0.09 | 99.75 | 0.35 | 0.67 |
| tbrstr36 lh | 0.00 | 48.24 | 20.72 | 0.23 | 13.74 | 3.05 | 14.12 | 0.29 | 0.03 | 0.02 | 0.11 | 100.55 | 0.35 | 0.61 |
| tbrstr36 lh | 0.16 | 48.54 | 20.85 | 0.16 | 13.80 | 2.64 | 14.24 | 0.26 | 0.12 | 0.00 | 0.15 | 100.92 | 0.35 | 0.61 |
| tbrstr36 lh | 0.03 | 47.96 | 20.58 | 0.22 | 13.23 | 3.76 | 14.44 | 0.28 | 0.11 | 0.02 | 0.08 | 100.72 | 0.34 | 0.61 |
| tbrstr36 lh | 0.03 | 48.10 | 20.64 | 0.17 | 12.99 | 3.56 | 14.56 | 0.27 | 0.08 | 0.00 | 0.09 | 100.51 | 0.33 | 0.61 |
| tbrstr36 lh | 0.02 | 47.87 | 20.53 | 0.20 | 13.04 | 3.58 | 14.47 | 0.22 | 0.07 | 0.01 | 0.08 | 100.10 | 0.34 | 0.61 |
| tbrstr36 lh | 0.00 | 48.30 | 20.70 | 0.17 | 13.48 | 3.32 | 14.24 | 0.24 | 0.14 | 0.02 | 0.10 | 100.71 | 0.35 | 0.61 |
| tbrstr36 lh | 0.00 | 47.83 | 20.49 | 0.18 | 13.27 | 3.73 | 14.31 | 0.23 | 0.04 | 0.03 | 0.09 | 100.21 | 0.34 | 0.61 |
| tbrstr36 lh | 0.09 | 47.84 | 20.50 | 0.28 | 13.42 | 3.72 | 14.46 | 0.23 | 0.02 | 0.02 | 0.06 | 100.64 | 0.34 | 0.61 |
| tbrstr36 lh | 0.07 | 48.29 | 20.68 | 0.24 | 13.70 | 3.33 | 14.30 | 0.28 | 0.08 | 0.01 | 0.08 | 101.06 | 0.35 | 0.61 |
| tbrstr36 lh | 0.07 | 47.90 | 20.43 | 0.22 | 13.72 | 3.52 | 14.06 | 0.29 | 0.13 | 0.01 | 0.08 | 100.43 | 0.35 | 0.61 |

lh = in lherzolite, hz = in harzburgite, and hz lh = in the contact harzburgite-lherzolite.

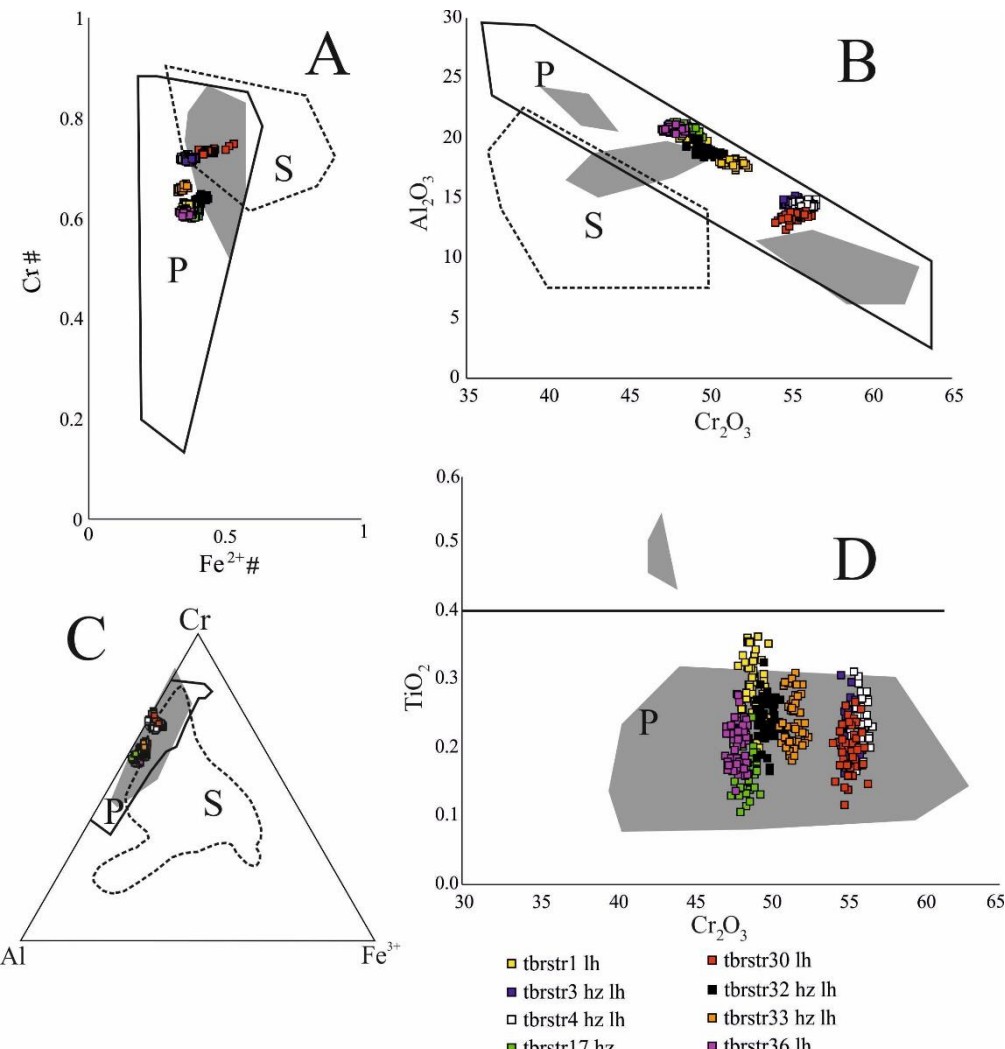

**Figure 6.** Composition of unaltered chromite from Kabaena Island chromitite: variation of the chromium number Cr# = Cr/(Cr + Al) and bivalent iron number $Fe^{2+}$# = $Fe^{2+}$/($Fe^{2+}$ + Mg) (**A**), correlation of $Al_2O_3$ (wt%) versus $Cr_2O_3$ (wt%) (**B**), Cr–Al–$Fe^{3+}$ atomic ratios (**C**), and variation of $Cr_2O_3$ (wt%) and $TiO_2$ (wt%) (**D**). Abbreviations: P = field of podiform chromitites and S = field of stratiform chromitites [16,36,37]. Grey field = Sulawesi chromitite [38].

In the diagram Cr# versus $TiO_2$, all the sample clusters from the field of MOR chromitite and some samples, characterized by a low content of Cr# also out from the field of boninitic chromitite (Figure 7B). Another group of analyses unequivocally plots the field of boninitic chromitite (Figure 7B). This diagram shows that the Kabaena chromitites are similar to those of Sulawesi, analyzed by Zaccarini et al. [38]. The relationships between Cr# and $Fe^{2+}$# suggest a similarity with the spinel formed in a SSZ and with those associated with the forearc peridotite. This diagram shows that the composition of the analyzed chromite is different from those reported from abyssal and back arc peridotite, as well as from spinel related to an oceanic plateau (Figure 7C). Our data differ from the disseminated spinels analyzed in ultramafic rocks of Kabaena ophiolite [10]. Figures 6 and 7 show that the analyzed chromitites form two small groups that plot separately. The groups do not correlate with a geographic position, as well as with the host peridotite.

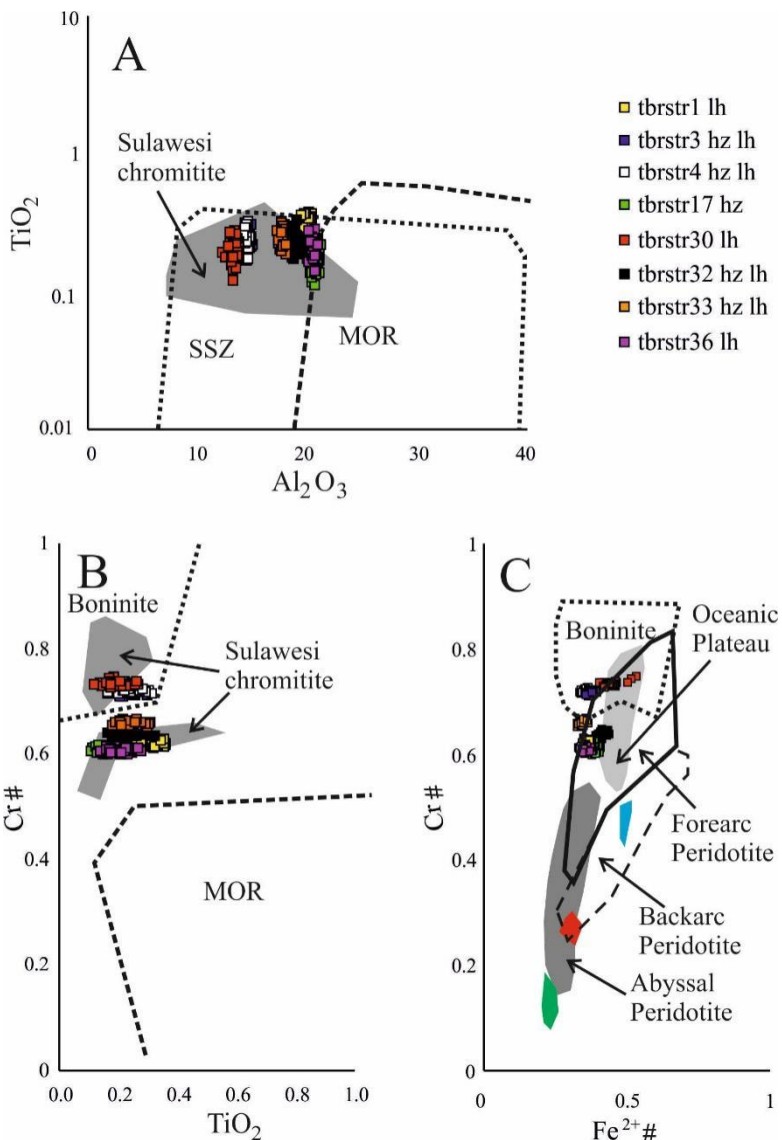

**Figure 7.** Composition of the chromitites from Kabaena Island compared with that from different the geodynamic setting and Sulawesi: $Al_2O_3$ (wt%) versus $TiO_2$ (wt%) (**A**), variation of $TiO_2$ (wt%) and Cr# = Cr/(Cr + Al) (**B**), and correlation between Cr# and $Fe^{2+}$# compared with the accessory spinel in the Kabaena lherzolite (green field), harzburgite (red field), and dunite (blue field) (**C**). Compositional fields from Kadarusman et al. [10], Dick and Bullen [18], Kamenetsky et al. [23], Arai [37], and Zaccarini et al. [38].

### 4.2. Silicates Texture and Composition

Most of the interstitial silicates in Kabaena Island chromitite have been totally altered in serpentine, talc and minor chlorite and actinolite. Only a few grains of preserved olivine and clinopyroxene have been found and analyzed (Tables 2 and 3). They occur as irregular grain in the matrix (Figure 5D) or as polygonal single-phase crystals (up to about 50 μm in size) enclosed in fresh chromite. Inclusions of olivine occur in samples tbrstr1 lh, tbrstr17 hz, tbrstr30 lh, tbrstr32 hz lh, and tbrstr36 lh, whereas preserved interstitial olivine has been found only in the tbrstr30 lh and tbrstr32 hz lh chromitite. Rare amphiboles have been found included in chromite and have been only qualitatively analyzed due to their small size. The composition of olivine has been plotted in the binary diagrams of Figure 8A–C, showing that olivine has a wide compositional range of Fo content.

**Table 2.** Selected microprobe analyses of olivine (wt%) from chromitites of the Kabaena Island ophiolite.

| Sample | SiO$_2$ | TiO$_2$ | Al$_2$O$_3$ | FeO | MnO | MgO | CaO | Na$_2$O | K$_2$O | Cr$_2$O$_3$ | CoO | NiO | Total | Fo% |
|---|---|---|---|---|---|---|---|---|---|---|---|---|---|---|
| | | | | | | Included | | | | | | | | |
| tbrstr1 lh | 42.64 | 0.00 | 0.00 | 3.88 | 0.01 | 53.28 | 0.03 | 0.00 | 0.00 | 0.18 | 0.00 | 0.72 | 100.75 | 96.07 |
| tbrstr1 lh | 42.59 | 0.05 | 0.01 | 3.93 | 0.10 | 53.45 | 0.02 | 0.01 | 0.00 | 0.23 | 0.02 | 0.64 | 101.05 | 96.04 |
| tbrstr1 lh | 42.76 | 0.03 | 0.02 | 3.53 | 0.05 | 53.74 | 0.01 | 0.00 | 0.02 | 0.64 | 0.00 | 0.65 | 101.44 | 96.45 |
| tbrstr1 lh | 42.79 | 0.00 | 0.02 | 3.72 | 0.10 | 53.21 | 0.02 | 0.01 | 0.00 | 0.27 | 0.00 | 0.69 | 100.81 | 96.22 |
| tbrstr1 lh | 42.47 | 0.04 | 0.02 | 3.64 | 0.09 | 53.58 | 0.01 | 0.00 | 0.00 | 0.26 | 0.00 | 0.84 | 100.94 | 96.33 |
| tbrstr1 lh | 42.90 | 0.00 | 0.01 | 3.39 | 0.13 | 54.01 | 0.04 | 0.00 | 0.00 | 0.69 | 0.00 | 0.79 | 101.97 | 96.60 |
| tbrstr1 lh | 42.92 | 0.00 | 0.00 | 3.42 | 0.05 | 53.82 | 0.05 | 0.00 | 0.02 | 0.69 | 0.01 | 0.73 | 101.69 | 96.56 |
| tbrstr1 lh | 42.63 | 0.02 | 0.01 | 3.75 | 0.06 | 53.92 | 0.01 | 0.00 | 0.01 | 0.72 | 0.02 | 0.71 | 101.85 | 96.25 |
| tbrstr1 lh | 42.40 | 0.02 | 0.00 | 3.66 | 0.01 | 53.67 | 0.00 | 0.00 | 0.02 | 0.74 | 0.00 | 0.67 | 101.18 | 96.31 |
| tbrstr1 lh | 42.06 | 0.03 | 0.04 | 3.62 | 0.05 | 54.29 | 0.01 | 0.00 | 0.00 | 0.74 | 0.01 | 0.78 | 101.63 | 96.40 |
| tbrstr1 lh | 42.70 | 0.00 | 0.02 | 4.29 | 0.01 | 53.55 | 0.02 | 0.00 | 0.00 | 0.45 | 0.00 | 0.59 | 101.65 | 95.70 |
| tbrstr1 lh | 42.96 | 0.01 | 0.00 | 4.08 | 0.04 | 53.79 | 0.03 | 0.00 | 0.00 | 0.43 | 0.05 | 0.65 | 102.05 | 95.92 |
| tbrstr17 lh | 42.71 | 0.03 | 0.01 | 4.02 | 0.07 | 53.72 | 0.02 | 0.00 | 0.00 | 0.51 | 0.02 | 0.84 | 101.95 | 95.97 |
| tbrstr17 lh | 40.97 | 0.01 | 0.00 | 4.27 | 0.07 | 53.73 | 0.00 | 0.00 | 0.01 | 1.01 | 0.01 | 0.66 | 100.74 | 95.73 |
| tbrstr17 lh | 41.10 | 0.07 | 0.01 | 4.09 | 0.07 | 53.86 | 0.01 | 0.00 | 0.00 | 0.86 | 0.00 | 0.75 | 100.82 | 95.91 |
| tbrstr17 lh | 41.82 | 0.00 | 0.00 | 4.35 | 0.02 | 53.53 | 0.01 | 0.00 | 0.00 | 0.81 | 0.00 | 0.65 | 101.19 | 95.64 |
| tbrstr17 lh | 42.42 | 0.00 | 0.01 | 4.88 | 0.07 | 52.84 | 0.02 | 0.00 | 0.00 | 0.42 | 0.03 | 0.61 | 101.30 | 95.07 |
| tbrstr30 lh | 42.68 | 0.00 | 0.04 | 4.93 | 0.03 | 52.46 | 0.03 | 0.01 | 0.00 | 0.61 | 0.06 | 0.74 | 101.58 | 95.00 |
| tbrstr30 lh | 42.38 | 0.03 | 0.00 | 4.85 | 0.05 | 52.41 | 0.02 | 0.00 | 0.00 | 0.57 | 0.02 | 0.81 | 101.14 | 95.07 |
| tbrstr30 lh | 42.03 | 0.00 | 0.00 | 5.13 | 0.00 | 52.27 | 0.04 | 0.00 | 0.00 | 0.77 | 0.02 | 0.63 | 100.90 | 94.78 |
| tbrstr32 hz lh | 42.21 | 0.00 | 0.01 | 5.25 | 0.09 | 53.00 | 0.05 | 0.01 | 0.00 | 0.40 | 0.00 | 0.49 | 101.50 | 94.74 |
| tbrstr32 hz lh | 42.43 | 0.00 | 0.00 | 4.79 | 0.08 | 52.67 | 0.07 | 0.01 | 0.01 | 0.54 | 0.00 | 0.50 | 101.09 | 95.14 |
| tbrstr32 hz lh | 41.19 | 0.00 | 0.01 | 4.59 | 0.07 | 52.85 | 0.05 | 0.00 | 0.00 | 0.40 | 0.00 | 0.62 | 99.78 | 95.36 |
| tbrstr32 hz lh | 42.66 | 0.01 | 0.00 | 4.91 | 0.04 | 52.60 | 0.07 | 0.00 | 0.01 | 0.09 | 0.02 | 0.53 | 100.93 | 95.02 |
| tbrstr32 hz lh | 42.23 | 0.05 | 0.01 | 4.37 | 0.07 | 53.09 | 0.05 | 0.02 | 0.00 | 0.60 | 0.03 | 0.58 | 101.09 | 95.59 |
| tbrstr32 hz lh | 42.66 | 0.02 | 0.01 | 4.09 | 0.01 | 53.02 | 0.04 | 0.00 | 0.00 | 0.59 | 0.07 | 0.57 | 101.08 | 95.85 |
| tbrstr36 lh | 42.74 | 0.01 | 0.00 | 3.92 | 0.05 | 52.94 | 0.00 | 0.01 | 0.00 | 0.58 | 0.00 | 0.64 | 100.88 | 96.02 |
| tbrstr36 lh | 42.62 | 0.02 | 0.01 | 3.82 | 0.02 | 52.70 | 0.04 | 0.00 | 0.00 | 0.63 | 0.01 | 0.74 | 100.59 | 96.09 |
| tbrstr36 lh | 42.37 | 0.00 | 0.01 | 3.75 | 0.01 | 52.95 | 0.00 | 0.00 | 0.02 | 0.85 | 0.00 | 0.80 | 100.75 | 96.18 |
| | | | | | | Intestitial | | | | | | | | |
| tbrstr30 lh | 42.29 | 0.03 | 0.00 | 5.39 | 0.11 | 52.15 | 0.02 | 0.01 | 0.00 | 0.16 | 0.00 | 0.77 | 100.92 | 94.52 |
| tbrstr30 lh | 42.90 | 0.02 | 0.02 | 5.05 | 0.07 | 52.07 | 0.01 | 0.00 | 0.00 | 0.04 | 0.00 | 0.62 | 100.80 | 94.84 |
| tbrstr30 lh | 42.65 | 0.00 | 0.01 | 5.14 | 0.16 | 52.15 | 0.02 | 0.00 | 0.00 | 0.04 | 0.05 | 0.69 | 100.90 | 94.76 |
| tbrstr30 lh | 42.41 | 0.02 | 0.00 | 5.02 | 0.10 | 51.91 | 0.02 | 0.00 | 0.00 | 0.02 | 0.04 | 0.57 | 100.12 | 94.85 |
| tbrstr30 lh | 42.79 | 0.02 | 0.00 | 5.10 | 0.10 | 52.17 | 0.03 | 0.00 | 0.00 | 0.04 | 0.00 | 0.71 | 100.95 | 94.80 |
| tbrstr30 lh | 42.27 | 0.02 | 0.00 | 5.24 | 0.08 | 51.99 | 0.02 | 0.00 | 0.00 | 0.08 | 0.00 | 0.60 | 100.29 | 94.65 |
| tbrstr30 lh | 42.45 | 0.02 | 0.04 | 5.16 | 0.15 | 52.28 | 0.02 | 0.00 | 0.01 | 0.13 | 0.00 | 0.73 | 100.98 | 94.75 |
| tbrstr30 lh | 42.86 | 0.00 | 0.02 | 5.58 | 0.11 | 52.38 | 0.03 | 0.00 | 0.00 | 0.04 | 0.02 | 0.71 | 101.74 | 94.36 |
| tbrstr30 lh | 42.32 | 0.00 | 0.00 | 5.26 | 0.14 | 51.92 | 0.04 | 0.01 | 0.00 | 0.03 | 0.00 | 0.51 | 100.23 | 94.62 |
| tbrstr30 lh | 42.80 | 0.02 | 0.01 | 5.31 | 0.16 | 52.14 | 0.03 | 0.00 | 0.00 | 0.04 | 0.03 | 0.52 | 101.05 | 94.60 |
| tbrstr30 lh | 42.94 | 0.01 | 0.01 | 5.08 | 0.06 | 51.72 | 0.02 | 0.00 | 0.02 | 0.04 | 0.00 | 0.60 | 100.49 | 94.77 |
| tbrstr30 lh | 42.66 | 0.00 | 0.03 | 5.07 | 0.07 | 52.14 | 0.03 | 0.00 | 0.00 | 0.02 | 0.00 | 0.61 | 100.62 | 94.83 |
| tbrstr30 lh | 43.29 | 0.02 | 0.00 | 5.15 | 0.13 | 52.39 | 0.03 | 0.00 | 0.00 | 0.02 | 0.00 | 0.74 | 101.77 | 94.78 |
| tbrstr32 hz lh | 42.37 | 0.01 | 0.00 | 6.26 | 0.14 | 51.64 | 0.04 | 0.01 | 0.00 | 0.03 | 0.01 | 0.49 | 100.99 | 93.64 |
| tbrstr32 hz lh | 42.29 | 0.02 | 0.04 | 5.59 | 0.08 | 51.91 | 0.03 | 0.00 | 0.00 | 0.01 | 0.00 | 0.49 | 100.46 | 94.31 |
| tbrstr32 hz lh | 42.22 | 0.04 | 0.02 | 5.43 | 0.08 | 51.24 | 0.02 | 0.00 | 0.00 | 0.03 | 0.00 | 0.42 | 99.51 | 94.39 |
| tbrstr32 hz lh | 42.32 | 0.04 | 0.01 | 5.42 | 0.06 | 51.47 | 0.03 | 0.01 | 0.00 | 0.02 | 0.03 | 0.62 | 100.00 | 94.42 |
| tbrstr32 hz lh | 42.53 | 0.00 | 0.01 | 5.73 | 0.09 | 52.11 | 0.04 | 0.02 | 0.01 | 0.12 | 0.00 | 0.65 | 101.32 | 94.19 |
| tbrstr32 hz lh | 41.92 | 0.01 | 0.01 | 5.70 | 0.09 | 51.90 | 0.04 | 0.00 | 0.02 | 0.05 | 0.00 | 0.57 | 100.32 | 94.19 |
| tbrstr32 hz lh | 42.52 | 0.00 | 0.00 | 5.50 | 0.13 | 51.69 | 0.05 | 0.00 | 0.00 | 0.07 | 0.02 | 0.49 | 100.46 | 94.37 |
| tbrstr32 hz lh | 41.92 | 0.06 | 0.02 | 5.54 | 0.06 | 51.32 | 0.05 | 0.00 | 0.00 | 0.01 | 0.03 | 0.58 | 99.60 | 94.29 |
| tbrstr32 hz lh | 41.73 | 0.01 | 0.03 | 5.80 | 0.13 | 51.57 | 0.05 | 0.00 | 0.01 | 0.02 | 0.00 | 0.48 | 99.81 | 94.07 |
| tbrstr32 hz lh | 41.87 | 0.00 | 0.00 | 5.56 | 0.11 | 51.58 | 0.04 | 0.00 | 0.00 | 0.01 | 0.05 | 0.70 | 99.92 | 94.29 |
| tbrstr32 hz lh | 41.95 | 0.00 | 0.05 | 5.43 | 0.10 | 51.79 | 0.12 | 0.00 | 0.00 | 0.02 | 0.00 | 0.60 | 100.06 | 94.45 |
| tbrstr32 hz lh | 41.92 | 0.00 | 0.01 | 5.46 | 0.05 | 51.72 | 0.04 | 0.00 | 0.00 | 0.03 | 0.06 | 0.56 | 99.84 | 94.40 |
| tbrstr32 hz lh | 42.37 | 0.00 | 0.01 | 5.64 | 0.07 | 51.55 | 0.04 | 0.00 | 0.01 | 0.02 | 0.02 | 0.60 | 100.32 | 94.21 |
| tbrstr32 hz lh | 41.89 | 0.00 | 0.00 | 5.66 | 0.06 | 51.52 | 0.05 | 0.01 | 0.01 | 0.05 | 0.01 | 0.48 | 99.73 | 94.20 |
| tbrstr32 hz lh | 42.46 | 0.00 | 0.00 | 5.51 | 0.11 | 51.24 | 0.05 | 0.00 | 0.00 | 0.00 | 0.04 | 0.55 | 99.95 | 94.31 |
| tbrstr32 hz lh | 42.01 | 0.00 | 0.02 | 5.72 | 0.07 | 51.72 | 0.07 | 0.01 | 0.00 | 0.05 | 0.00 | 0.53 | 100.19 | 94.16 |
| tbrstr32 hz lh | 42.42 | 0.04 | 0.01 | 5.63 | 0.04 | 51.78 | 0.04 | 0.00 | 0.01 | 0.04 | 0.08 | 0.49 | 100.59 | 94.25 |

lh = in lherzolite, hz = in harzburgite, and hz lh = in contact harzburgite-lherzolite.

**Table 3.** Selected microprobe analyses of clinopyroxene (wt%) from chromitites of Kabaena Island ophiolite.

| Sample | SiO$_2$ | TiO$_2$ | Al$_2$O$_3$ | FeO | MnO | MgO | CaO | Na$_2$O | K$_2$O | Cr$_2$O$_3$ | NiO | Total |
|---|---|---|---|---|---|---|---|---|---|---|---|---|
| | | | | | Included | | | | | | | |
| tbrstr1 lh | 54.02 | 0.22 | 1.97 | 1.48 | 0.05 | 17.37 | 23.53 | 0.12 | 0.00 | 1.52 | 0.00 | 100.28 |
| tbrstr1 lh | 54.37 | 0.20 | 2.03 | 1.46 | 0.05 | 17.31 | 23.75 | 0.11 | 0.01 | 1.55 | 0.09 | 100.92 |
| tbrstr1 lh | 55.03 | 0.17 | 1.40 | 1.18 | 0.09 | 17.54 | 24.02 | 0.12 | 0.01 | 1.64 | 0.00 | 101.19 |
| tbrstr1 lh | 53.96 | 0.08 | 0.92 | 1.25 | 0.01 | 18.38 | 23.33 | 0.16 | 0.01 | 1.14 | 0.04 | 99.26 |
| | | | | | Interstitial | | | | | | | |
| tbrstr3 hz lh | 53.80 | 0.08 | 0.85 | 1.47 | 0.01 | 18.67 | 23.77 | 0.19 | 0.00 | 1.31 | 0.00 | 100.14 |
| tbrstr3 hz lh | 53.46 | 0.11 | 1.12 | 1.41 | 0.01 | 18.56 | 23.13 | 0.17 | 0.02 | 1.07 | 0.12 | 99.17 |
| tbrstr3 hz lh | 53.40 | 0.08 | 0.75 | 1.29 | 0.08 | 18.70 | 23.60 | 0.13 | 0.01 | 1.01 | 0.16 | 99.22 |
| tbrstr4 hz lh | 52.59 | 0.12 | 1.18 | 1.18 | 0.04 | 17.93 | 23.81 | 0.49 | 0.00 | 1.98 | 0.02 | 99.35 |
| tbrstr17 hz | 54.23 | 0.06 | 1.70 | 1.81 | 0.05 | 17.94 | 22.48 | 0.27 | 0.00 | 1.30 | 0.00 | 99.83 |
| tbrstr17 hz | 54.71 | 0.11 | 1.53 | 1.65 | 0.01 | 17.57 | 23.40 | 0.25 | 0.00 | 1.45 | 0.07 | 100.73 |
| tbrstr30 lh | 54.04 | 0.04 | 1.41 | 2.01 | 0.12 | 18.62 | 22.32 | 0.19 | 0.00 | 1.17 | 0.06 | 100.00 |
| tbrstr30 lh | 55.32 | 0.11 | 1.49 | 1.85 | 0.08 | 17.52 | 23.25 | 0.24 | 0.00 | 1.25 | 0.02 | 101.12 |
| tbrstr30 lh | 55.17 | 0.08 | 1.37 | 1.85 | 0.05 | 18.41 | 22.29 | 0.17 | 0.00 | 1.23 | 0.02 | 100.66 |
| tbrstr30 lh | 55.14 | 0.08 | 1.27 | 1.91 | 0.11 | 18.16 | 22.54 | 0.18 | 0.03 | 1.28 | 0.14 | 100.88 |
| tbrstr30 lh | 55.89 | 0.01 | 1.42 | 1.94 | 0.16 | 17.82 | 22.94 | 0.19 | 0.00 | 1.29 | 0.07 | 101.74 |
| tbrstr30 lh | 54.90 | 0.07 | 1.46 | 1.97 | 0.07 | 17.87 | 22.76 | 0.26 | 0.00 | 1.38 | 0.09 | 100.83 |
| tbrstr30 lh | 54.97 | 0.05 | 1.40 | 1.92 | 0.06 | 17.91 | 22.74 | 0.20 | 0.00 | 1.22 | 0.01 | 100.47 |
| tbrstr30 lh | 55.26 | 0.09 | 1.32 | 1.97 | 0.10 | 18.23 | 22.19 | 0.16 | 0.00 | 1.28 | 0.11 | 100.70 |
| tbrstr30 lh | 54.91 | 0.01 | 1.35 | 2.15 | 0.04 | 18.70 | 21.55 | 0.19 | 0.02 | 1.32 | 0.18 | 100.43 |
| tbrstr30 lh | 55.12 | 0.09 | 1.11 | 1.78 | 0.10 | 18.48 | 22.07 | 0.19 | 0.01 | 1.10 | 0.00 | 100.03 |
| tbrstr30 lh | 55.33 | 0.05 | 1.08 | 1.58 | 0.07 | 18.10 | 22.71 | 0.18 | 0.00 | 1.11 | 0.06 | 100.26 |
| tbrstr32 hz lh | 54.61 | 0.12 | 1.88 | 1.91 | 0.12 | 17.97 | 22.23 | 0.30 | 0.00 | 1.56 | 0.04 | 100.74 |
| tbrstr32 hz lh | 54.33 | 0.15 | 1.86 | 1.85 | 0.07 | 17.68 | 22.56 | 0.21 | 0.01 | 1.56 | 0.00 | 100.28 |
| tbrstr32 hz lh | 54.35 | 0.17 | 1.79 | 1.78 | 0.05 | 17.82 | 22.78 | 0.19 | 0.01 | 1.23 | 0.18 | 100.36 |
| tbrstr32 hz lh | 54.00 | 0.14 | 2.10 | 2.02 | 0.06 | 18.19 | 22.02 | 0.23 | 0.01 | 1.38 | 0.08 | 100.24 |
| tbrstr32 hz lh | 54.55 | 0.17 | 2.03 | 1.91 | 0.00 | 17.94 | 22.23 | 0.29 | 0.02 | 1.26 | 0.00 | 100.42 |
| tbrstr32 hz lh | 54.11 | 0.13 | 1.89 | 1.81 | 0.07 | 17.96 | 23.05 | 0.35 | 0.01 | 1.27 | 0.02 | 100.67 |
| tbrstr32 hz lh | 54.83 | 0.12 | 1.89 | 1.86 | 0.09 | 17.88 | 22.16 | 0.36 | 0.01 | 1.17 | 0.15 | 100.54 |
| tbrstr32 hz lh | 54.63 | 0.17 | 1.84 | 1.71 | 0.08 | 17.44 | 22.76 | 0.34 | 0.00 | 1.26 | 0.00 | 100.23 |
| tbrstr32 hz lh | 54.34 | 0.11 | 2.12 | 1.90 | 0.04 | 17.30 | 22.41 | 0.35 | 0.01 | 1.30 | 0.01 | 99.91 |
| tbrstr32 hz lh | 53.38 | 0.11 | 2.33 | 1.83 | 0.07 | 18.10 | 21.83 | 0.28 | 0.01 | 1.26 | 0.06 | 99.27 |
| tbrstr32 hz lh | 54.34 | 0.15 | 1.86 | 1.86 | 0.11 | 17.72 | 22.52 | 0.34 | 0.00 | 1.42 | 0.00 | 100.31 |
| tbrstr32 hz lh | 54.36 | 0.15 | 1.80 | 1.79 | 0.11 | 17.70 | 22.76 | 0.35 | 0.00 | 1.48 | 0.05 | 100.53 |
| tbrstr32 hz lh | 53.87 | 0.14 | 1.88 | 1.73 | 0.09 | 17.62 | 22.63 | 0.30 | 0.01 | 1.41 | 0.00 | 99.72 |
| tbrstr32 hz lh | 54.35 | 0.14 | 1.86 | 1.65 | 0.05 | 17.59 | 22.58 | 0.35 | 0.00 | 1.42 | 0.16 | 100.16 |
| tbrstr32 hz lh | 54.10 | 0.08 | 2.18 | 1.76 | 0.06 | 18.13 | 22.21 | 0.28 | 0.00 | 1.29 | 0.01 | 100.10 |
| tbrstr32 hz lh | 54.06 | 0.13 | 2.08 | 1.94 | 0.07 | 17.52 | 22.35 | 0.33 | 0.00 | 1.37 | 0.12 | 99.99 |
| tbrstr32 hz lh | 54.73 | 0.15 | 2.06 | 2.10 | 0.05 | 18.48 | 21.77 | 0.26 | 0.00 | 1.30 | 0.03 | 100.91 |
| tbrstr32 hz lh | 54.72 | 0.11 | 1.98 | 1.90 | 0.09 | 17.70 | 22.18 | 0.31 | 0.00 | 1.21 | 0.01 | 100.20 |
| tbrstr32 hz lh | 54.34 | 0.11 | 1.90 | 1.91 | 0.11 | 17.87 | 21.86 | 0.34 | 0.00 | 1.16 | 0.03 | 99.62 |
| tbrstr32 hz lh | 54.35 | 0.15 | 1.90 | 1.99 | 0.09 | 17.71 | 21.89 | 0.31 | 0.00 | 1.19 | 0.04 | 99.61 |
| tbrstr32 hz lh | 54.56 | 0.08 | 1.78 | 1.70 | 0.10 | 17.69 | 22.49 | 0.35 | 0.00 | 1.25 | 0.04 | 100.04 |
| tbrstr32 hz lh | 54.65 | 0.13 | 1.90 | 2.09 | 0.06 | 17.87 | 22.10 | 0.34 | 0.00 | 1.40 | 0.11 | 100.65 |
| tbrstr32 hz lh | 54.37 | 0.16 | 2.10 | 2.17 | 0.05 | 18.42 | 21.68 | 0.28 | 0.01 | 1.27 | 0.11 | 100.61 |
| tbrstr32 hz lh | 54.25 | 0.15 | 1.97 | 1.72 | 0.09 | 17.38 | 22.96 | 0.30 | 0.01 | 1.50 | 0.06 | 100.39 |
| tbrstr32 hz lh | 54.66 | 0.18 | 1.73 | 1.47 | 0.10 | 17.47 | 23.40 | 0.29 | 0.00 | 1.12 | 0.09 | 100.51 |
| tbrstr32 hz lh | 54.62 | 0.14 | 1.95 | 1.83 | 0.10 | 17.92 | 22.56 | 0.31 | 0.00 | 1.29 | 0.04 | 100.76 |
| tbrstr33 hz lh | 54.05 | 0.18 | 1.54 | 1.75 | 0.03 | 17.78 | 22.81 | 0.23 | 0.00 | 1.63 | 0.03 | 100.02 |
| tbrstr33 hz lh | 54.11 | 0.08 | 1.56 | 1.69 | 0.13 | 18.00 | 23.04 | 0.25 | 0.00 | 1.60 | 0.03 | 100.50 |
| tbrstr33 hz lh | 54.73 | 0.12 | 1.54 | 1.54 | 0.05 | 17.78 | 21.68 | 0.20 | 0.01 | 1.52 | 0.18 | 99.39 |

lh = in lherzolite, hz = in harzburgite, and hz lh = in the contact harzburgite-lherzolite.

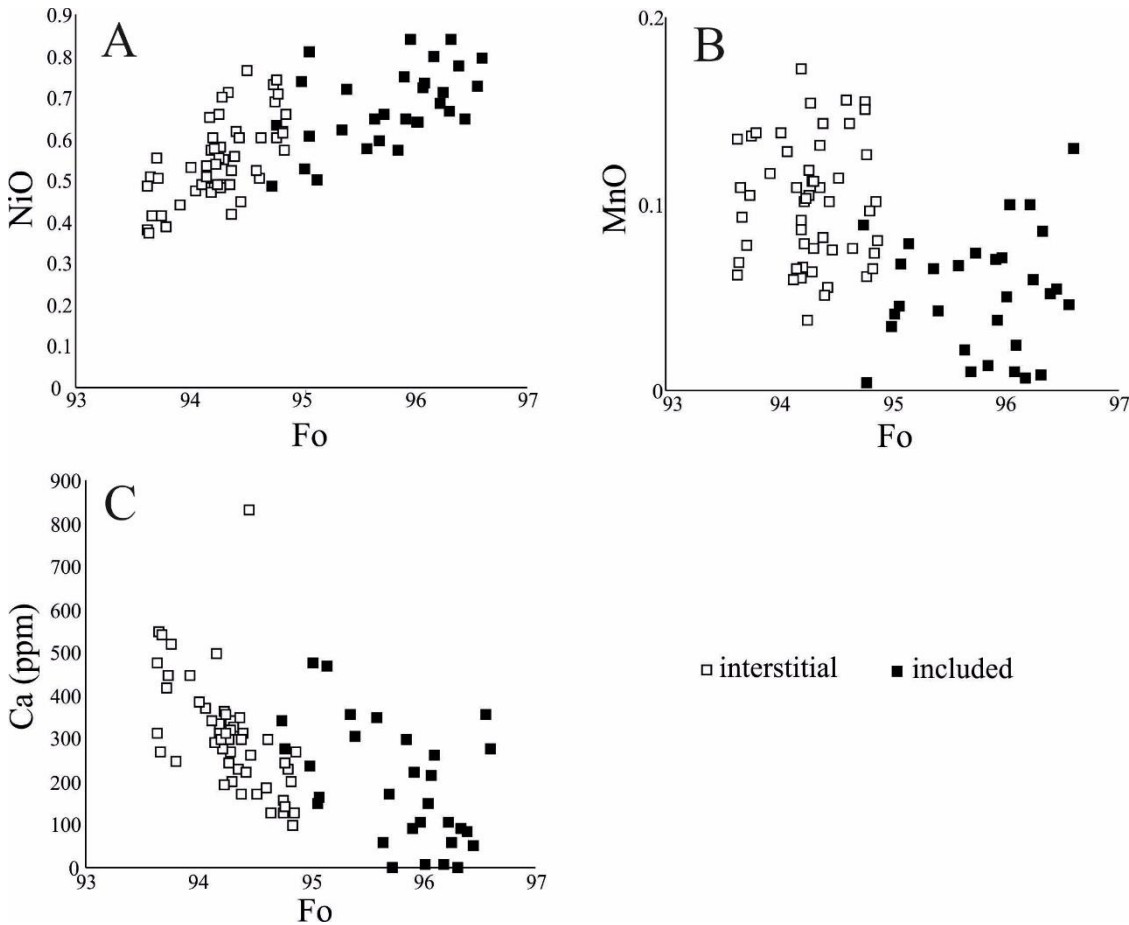

**Figure 8.** Compositional variation of NiO (wt%) (**A**), MnO (wt%) (**B**), and Ca (**C**) as a function of forsterite molar % in the olivine from Kabaena Island chromitites.

In particular, the Fo content of interstitial olivine is comprised between 93.64 and 94.85 and those of the included olivine between 94.74 and 96.60 (Figure 8A–C and Table 2). The analyzed trace elements, i.e., Ni, Mn, and Ca, display a similar amount in both the interstitial and included olivine (Figure 8A–C). However, a weak enrichment in MnO and NiO is observed in the interstitial and included olivine, respectively, as shown in Figure 8A,B. The low content of Ca in ppm clearly suggests a mantle origin for the analyzed olivine (Figure 8C). NiO slightly correlates positively with the Fo content (Figure 8A); on the contrary, CaO was negatively associated (Figure 8C). Clinopyroxene is abundant as the interstitial phase occurring in chromitite tbrstr3 hz lh, tbrstr4 hz lh, tbrstr17 hz, tbrstr30 lh, tbrstr32 hz lh, and tbrstr33 hz lh and is rare as an inclusion in fresh chromite crystal and has been found only in sample tbrstr1 lh. The composition of clinopyroxene plotted in the ternary diagram proposed by Morimoto [39] shows that most of the analyzed clinopyroxene falls within the diopside field (Figure 9A). Only one sample (tbrstr30 hz lh) can be frankly classified as augite, and few analyses of the samples tbrstr32 hz lh and tbrstr33 hz lh fall around the contact field between augite and diopside (Figure 9A). The analyzed clinopyroxenes contain $Al_2O_3$ (up to 2.33 wt%) and $Cr_2O_3$ (up to 1.98 wt%) and very low amounts of MnO, NiO, $TiO_2$, $Na_2O$, and $K_2O$ (Table 3). According to the binary diagrams of Rogkala et al. [40] and Lian et al. [41] that take into consideration the variations of $Al_2O_3$ and $Cr_2O_3$ versus the Mg# in clinopyroxene to distinguish the abyssal from the forearc peridotite, Kabaena Island clinopyroxenes are very similar to those formed in the forearc, although showing an enrichment in $Cr_2O_3$, probably due to the influence of the chromite matrix (Figure 9B,C).

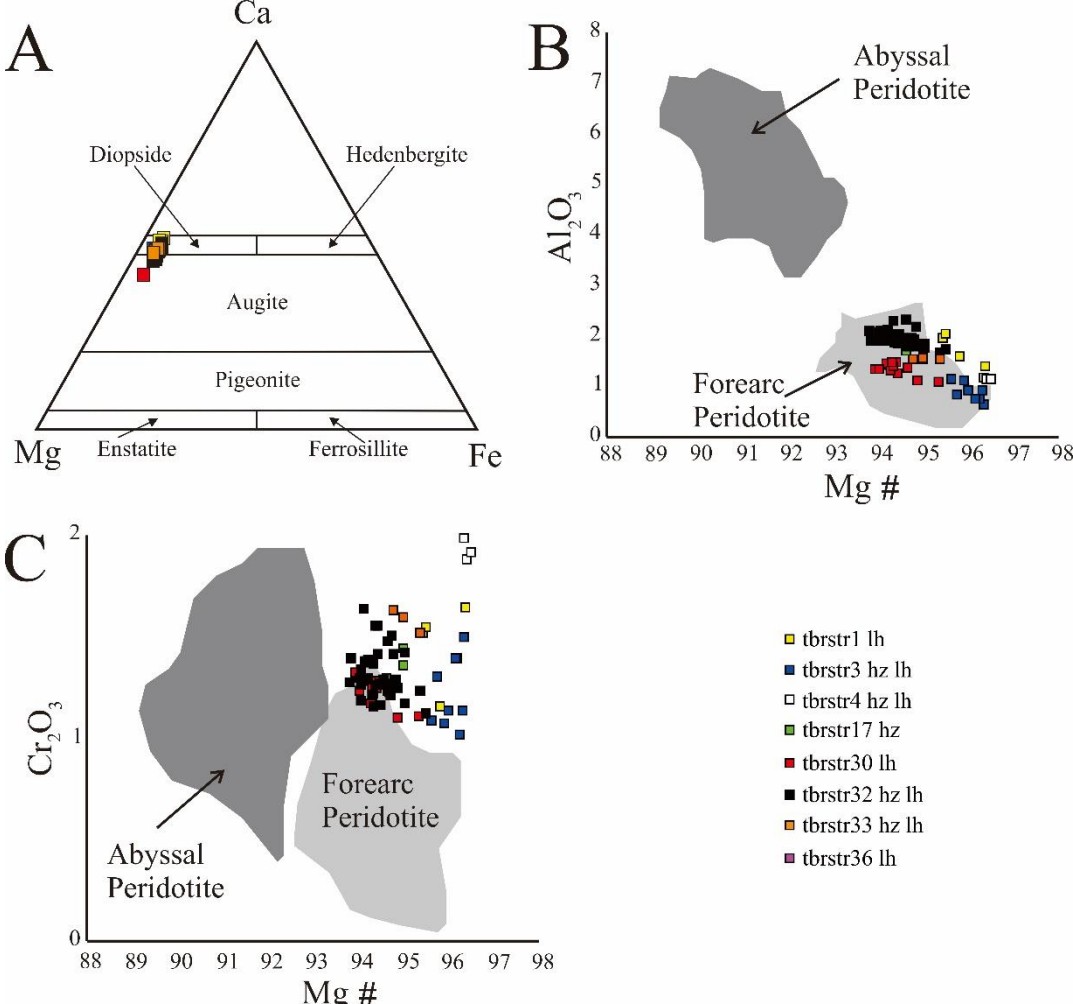

**Figure 9.** Composition of clinopyroxene from Kabaena Island chromitite: classification in the ternary diagram of Morimoto [39] (**A**), correlation between $Al_2O_3$ (wt%) versus Mg# = Mg/(Mg + $Fe^{2+}$) (**B**), and variation of $Cr_2O_3$ (wt%) and Mg# (**C**). Compositional fields from Rogkala et al. [40] and Lian et al. [41].

*4.3. Platinum Group Minerals (PGM) and Other Accessory Phases*

PGM are very rare in Kabaena Island chromitites and have been encountered only in the tbrstr30 lh and tbrstr32 hz lh samples. Based on their chemical compositions, they consist only of Os, Ir, and Ru phases, as typical for the mantle hosted podiform chromitite. In particular, the following PGM have been recognized, i.e., laurite and Os-Ir alloy. Laurite is the most abundant PGM, and it occurs as small grains (less than 10 μm in size) in the following textural positions: included in fresh chromite crystals and in contact with microfractures of chromite crystals (Figure 10A–C).

Few grains, enclosed in fresh chromite, are associated with silicates, such as clinopyroxene and amphibole (Figure 10C). Os-Ir alloy forms small blebs, about 1 μm in size, in a grain of awaruite that occurs in the altered silicate matrix in contact with serpentine (Figure 11A,B). Due to the small size of the encountered PGM, only few laurite grains have been quantitatively analyzed, and the results have been plotted in the Ru-Os-Ir ternary diagram of Figure 12 and compared with laurite analyzed in Sulawesi chromitite [38]. According to this diagram, both the analyzed laurite, i.e., included in fresh chromite or in contact with microfracture, displays a similar composition.

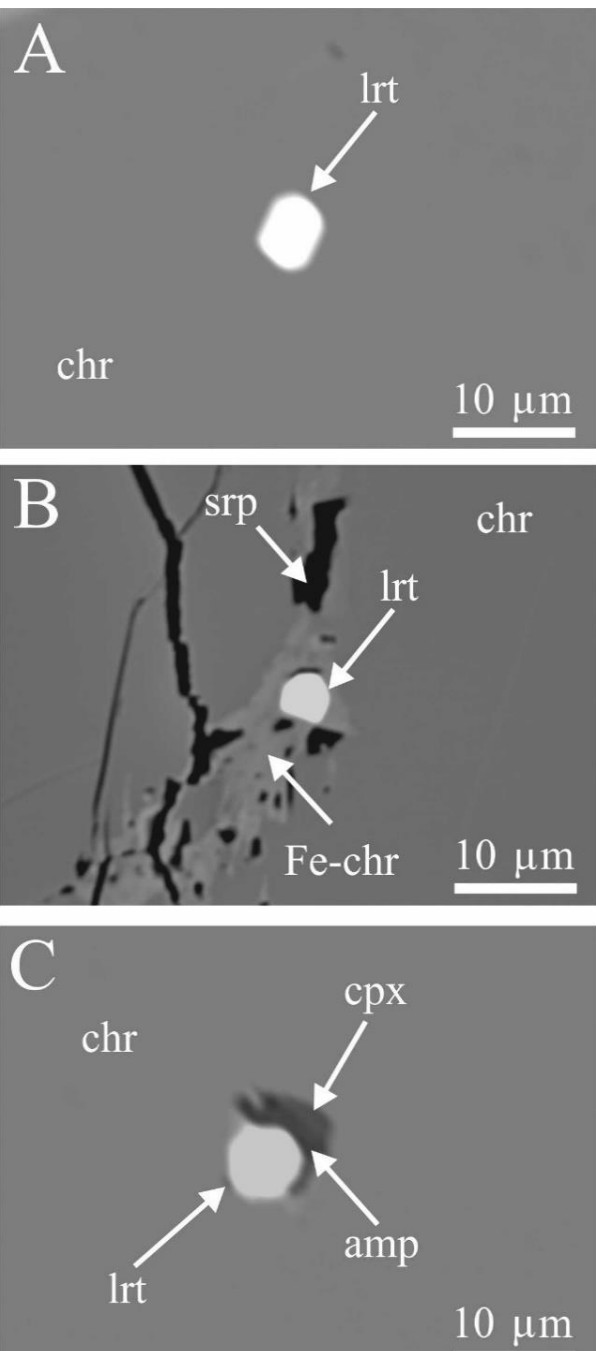

**Figure 10.** BSE images of magmatic laurite from Kabaena Island chromitite: single-phase enclosed in fresh chromite (**A**), single phase in contact with ferrian chromite (**B**), and the inclusion of laurite and silicates (**C**). Abbreviations: lrt = laurite, chr = chromite, srp = serpentine, Fe-chr = ferrian chromite, cpx = clinopyroxene, and amp = amphibole.

They differ only because the laurite enclosed in the chromite from Kabaena shows a slight enrichment in Os and Pd compared to those occurring in the microfracture (Table 4). The Rh content is always low, being comprised of between 0.62 and 1.05 wt% (Table 4). Rare and small inclusions (less than 10 μm in size) of sulfides have also been qualitatively analyzed. They consist of chalcopyrite, millerite, and pentlandite. Small spots of awaruite, analyzed by EDS, have also been recognized in the silicate matrix of the investigated chromites.

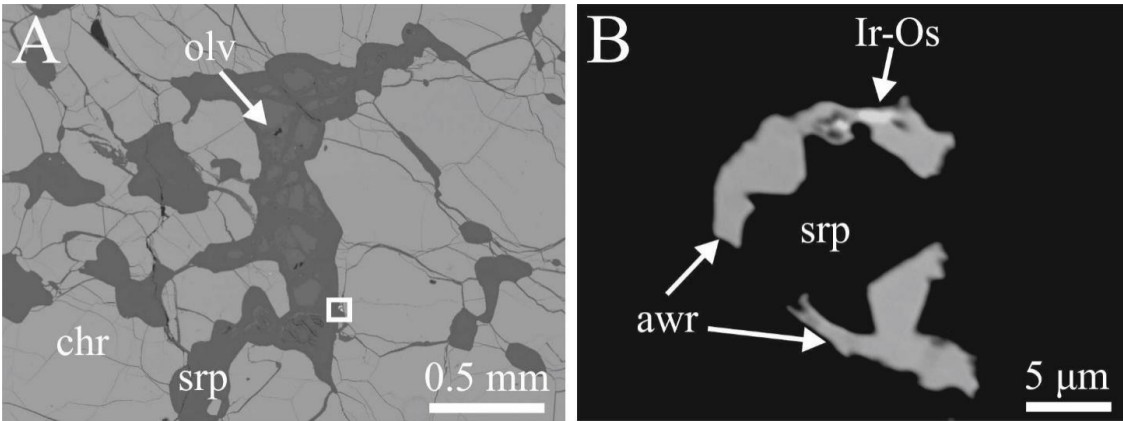

**Figure 11.** BSE images of secondary Ir-Os alloy from Kabaena Island chromitite (**A**) and enlargement of A (**B**). Abbreviations: chr = chromite, srp = serpentine, olv = olivine, Ir-Os = iridium–osmium alloy, and awr = awaruite.

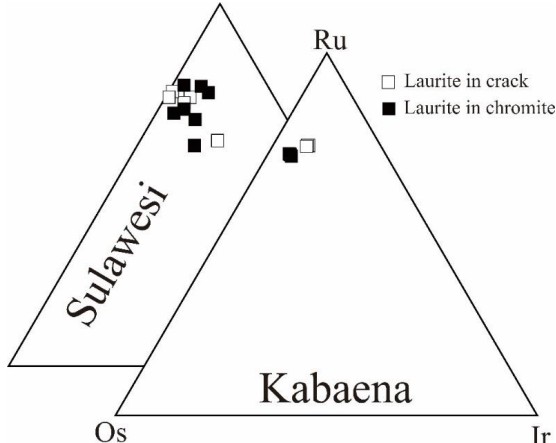

**Figure 12.** Plots (atom %) of the compositions of laurite from the chromitites of Kabaena (present work) and Sulawesi [38].

**Table 4.** Selected microprobe analyses of laurite (wt% and at%) from chromitites of the Kabaena Island ophiolite.

| Sample | As | S | Rh | Ir | Os | Pd | Ru | Total |
|---|---|---|---|---|---|---|---|---|
| | | | | in crack | | | | |
| tbrstr32 hz lh | 0.60 | 34.01 | 0.96 | 8.21 | 16.21 | 1.77 | 38.08 | 99.83 |
| tbrstr32 hz lh | 0.65 | 33.63 | 0.84 | 8.54 | 15.97 | 1.82 | 38.11 | 99.55 |
| tbrstr32 hz lh | 0.72 | 34.83 | 0.92 | 8.44 | 16.10 | 1.59 | 38.23 | 100.84 |
| tbrstr32 hz lh | 0.56 | 34.17 | 1.05 | 8.58 | 15.83 | 2.03 | 38.28 | 100.51 |
| | | | | Included | | | | |
| tbrstr32 hz lh | 0.39 | 33.10 | 0.75 | 5.43 | 22.09 | 0.63 | 36.09 | 98.47 |
| tbrstr32 hz lh | 0.40 | 33.32 | 0.70 | 5.16 | 21.83 | 0.77 | 36.04 | 98.21 |
| tbrstr32 hz lh | 0.46 | 33.53 | 0.88 | 5.42 | 21.79 | 0.65 | 36.15 | 98.88 |
| tbrstr32 hz lh | 0.47 | 34.30 | 0.88 | 5.69 | 22.52 | 0.37 | 36.78 | 101.01 |
| tbrstr32 hz lh | 0.48 | 33.92 | 0.62 | 6.16 | 22.39 | 0.35 | 36.80 | 100.73 |
| tbrstr32 hz lh | 0.41 | 34.10 | 0.67 | 5.62 | 22.82 | 0.06 | 36.68 | 100.36 |
| tbrstr32 hz lh | 0.40 | 34.04 | 0.77 | 6.02 | 22.65 | 0.63 | 36.45 | 100.95 |
| Comment | As | S | Rh | Ir | Os | Pd | Ru | Total |
| | | | | in crack | | | | |
| tbrstr32 hz lh | 0.50 | 66.32 | 0.58 | 2.67 | 5.33 | 1.04 | 23.56 | 100 |
| tbrstr32 hz lh | 0.54 | 66.04 | 0.51 | 2.80 | 5.29 | 1.08 | 23.74 | 100 |
| tbrstr32 hz lh | 0.59 | 66.78 | 0.55 | 2.70 | 5.20 | 0.92 | 23.25 | 100 |
| tbrstr32 hz lh | 0.46 | 66.23 | 0.63 | 2.77 | 5.17 | 1.19 | 23.54 | 100 |

**Table 4.** *Cont.*

| Sample | As | S | Rh | Ir | Os | Pd | Ru | Total |
|---|---|---|---|---|---|---|---|---|
| | | | | Included | | | | |
| tbrstr32 hz lh | 0.33 | 66.51 | 0.47 | 1.82 | 7.48 | 0.38 | 23.00 | 100 |
| tbrstr32 hz lh | 0.34 | 66.76 | 0.44 | 1.72 | 7.37 | 0.46 | 22.90 | 100 |
| tbrstr32 hz lh | 0.39 | 66.74 | 0.54 | 1.80 | 7.31 | 0.39 | 22.83 | 100 |
| tbrstr32 hz lh | 0.39 | 66.86 | 0.53 | 1.85 | 7.40 | 0.22 | 22.75 | 100 |
| tbrstr32 hz lh | 0.40 | 66.64 | 0.38 | 2.02 | 7.41 | 0.21 | 22.93 | 100 |
| tbrstr32 hz lh | 0.34 | 66.96 | 0.41 | 1.84 | 7.55 | 0.03 | 22.85 | 100 |
| tbrstr32 hz lh | 0.33 | 66.72 | 0.47 | 1.97 | 7.48 | 0.37 | 22.66 | 100 |

hz lh = in the contact harzburgite-lherzolite.

### 4.4. Composition of the Chromitite Parental Melt

It is widely accepted that the chromite composition is strictly related to the nature of the parental melt from which its host chromitite precipitated, and several equations have been proposed to calculate the $Al_2O_3$, $TiO_2$, and the FeO/MgO ratio of the chromitite parental melt [42,43]. Therefore, using the empirical approach of Maurel and Maurel [42] and Rollinson [43], we calculated the composition of the melt parental to the chromitites of Kabaena Island in terms of the $Al_2O_3$ and $TiO_2$ contents, as well as the FeO/MgO ratio. The data, plotted in the diagrams of Figure 13A–E, show that the obtained results are controversial.

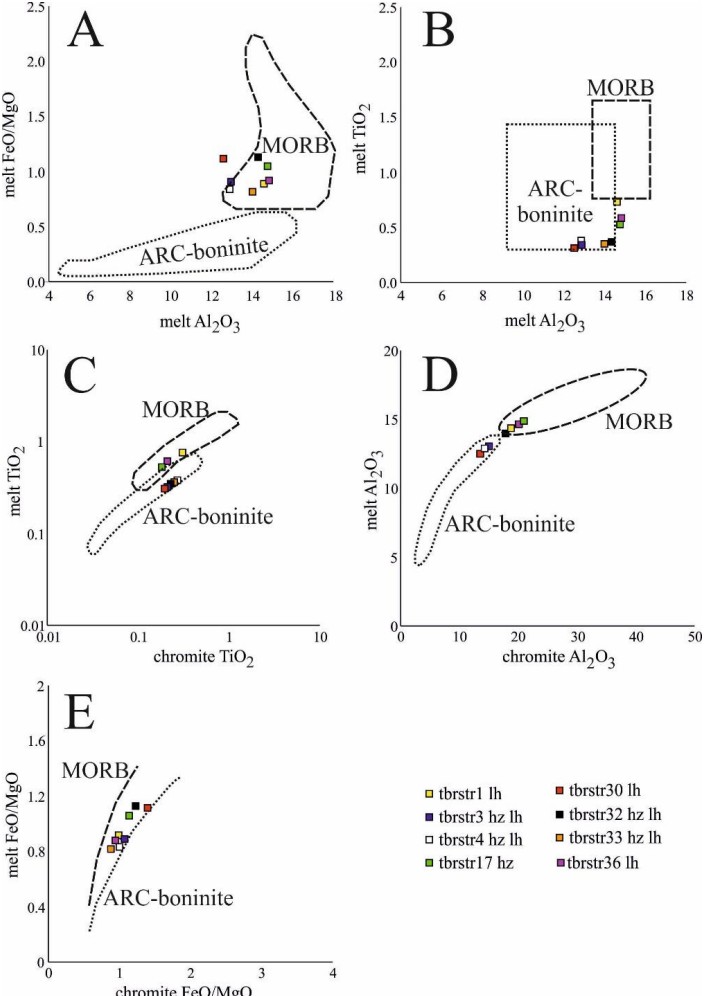

**Figure 13.** Chromite composition–melt relationships in Kabaena Island chromitites: $Al_2O_3$ versus FeO/MgO melt (**A**), melt variation in terms of $Al_2O_3$ and $TiO_2$ (**B**), correlation between $TiO_2$ (wt%) in chromite and melt (**C**), $Al_2O_3$ (wt%) and $Al_2O_3$ in the melt (**D**), and the ratio FeO/MgO in the chromite and in the melt (**E**).

The variations between the FeO/MgO ratio versus $Al_2O_3$ in the melt are consistent with a MORB origin (Figure 13A). On the contrary, the correlation $TiO_2$ versus $Al_2O_3$ resembles those related with arc-boninite melts (Figure 13B). The calculated amount of $Al_2O_3$ and $TiO_2$ in the melts plotted versus the $Al_2O_3$ and $TiO_2$ concentrations of the host chromite reveals a bimodal character of Kabaena Island chromitite, since a group of samples falls in the MORB field and another group in those of arc (Figure 13C,D). The data in the binary diagram FeO/MgO ratio in the melt and the FeO/MgO ratio in the chromite plot in a narrow area comprised between MORB and the arc (Figure 13E).

## 5. Discussion

### 5.1. Chromite, Silicates, and Parental Melt Composition: Geodynamic Setting Implications for the Chromitite Formation

Although the East Sulawesi ophiolite is numbered among the largest ophiolites in the world, it hosts a limited number of studied chromitite occurrences. Only recently, Zaccarini et al. [38] and Septiana et al. [32] reported the presence of several small chromite deposits from the South and Southeast Arms of Sulawesi.

The Sulawesi chromitites consist of Cr-rich (Cr# up to 0.85) to Al-rich (minimum Cr# = 0.54), and formed is an SSZ, due to the fractionation of a single batch of magma with an initial boninitic composition during its ascent. The model for the genesis of the Sulawesi chromitites suggested the accumulation of Cr-rich chromitites at deep mantle levels and the formation of the Al-rich chromitites close to or above the Moho-transition zone [38]. The Cr# of Kabaena Island chromitites is comprised between 0.65 and 0.75, indicating the absence of Al-rich chromitites (Figure 6A).

It is widely accepted that the formation of chromitite in the mantle sequence of ophiolite is controlled by several factors, including the composition of the parent magma, the degree of partial melting of the mantle source, spreading rate, the presence of metasomatic fluids, and the geodynamic environment [21,23,44–51]. Two types of ophiolites have been recognized: (1) the lherzolite type (LOT), which represents the products of low degrees of melting of a fertile mantle, and (2) the harzburgite type (HOT), which corresponds to high degrees of melting of a depleted mantle [52]. Large chromite deposits have been preferentially described in the mantle sections of the HOT ophiolite.

Kabaena Island chromitites are hosted in a mantle peridotite composed of both harzburgite and lherzolite; therefore, this is probably the reason why they form small, mineralized bodies and not huge deposits. The chromite composition can be used to better understand the origin of its host chromitites. In particular, Cr-rich chromitite results from the melting of a highly depleted mantle producing picritic and boninitic magmas formed in a SSZ, and Al-rich chromitite associated with magmas corresponding to a lower degree of partial melting in a MOR environment.

According to the chromite composition, the studied chromitites of Kabaena Island can be frankly classified as podiform. They display similarity to those generated in an SSZ, being poor in $TiO_2$ and $Al_2O_3$ and having a Cr# higher than 0.65 (Figures 6A and 7A). In particular, the chromite composition, in terms of Fe# versus Cr# suggest that they formed in the forearc portion of the SSZ (Figure 7C).

The degree of partial melting was not an important key factor during the precipitation of the studied chromitites, since they display the same composition when they are hosted in the lherzolite or in the harzburgite. Additionally, the composition of the analyzed clinopyroxene is consistent with an origin of Kabaena Island chromitite in a forearc ophiolite (Figure 9B,C).

The analyzed interstitial and included olivine (Figure 8A–C) are characterized by values of forsterite, NiO, MnO, and Ca similar to those reported from ophiolitic mantle-hosted chromitites [53]. Therefore, we can argue that the studied olivine and its host chromitites formed in the mantle section of Kabaena Island ophiolite. However, as previously noticed by Garuti et al. [53] and Zaccarini et al. [38], in the olivine associated with the Urals and Sulawesi chromitites, the olivine of Kabaena Island enclosed in chromite are enriched in

forsterite compared to those occurring interstitially. This feature can be attributed to the adjustment of the olivine–chromite Mg/Fe ratios in the post-magmatic stage representing the closure temperature for the Fe and Mg exchange reaction due to the chromite/olivine mass ratio coupled with the cooling and diffusion rate [17,53–59]. Alternatively, the high content of MgO in the olivine enclosed in the chromite is due to the fact that it precipitated before the host chromite and the intestinal olivine. As a consequence, the inclusions of olivine have a more refractory character and are less evolved compared to those that are interstitial.

In several podiform chromitites formed in an SSZ, the hydrous conditions during their crystallization are recorded by the presence of abundant hydrous silicates such as amphibole and phlogopite that occur as the primary inclusion in the chromite crystals [46,60,61]. The most abundant silicate inclusions in Kabaena Island chromitites are olivine and clinopyroxene, suggesting the presence of a magma poor in hydrous fluids during the precipitation. Therefore, the metasomatic reaction with the mantle peridotite was limited, giving rise to small size deposits such as those observed in Kabaena Island. The mineral chemistry of the analyzed chromite, clinopyroxene, and olivine suggest that Kabaena Island chromitites formed in the mantle section of East Sulawesi ophiolite in an SSZ geodynamic setting in agreement with the model proposed by Monnier et al. [8].

However, if we consider the calculated melt composition from which Kabaena Island chromitite precipitated, the situation becomes more complex, being compatible with both a MORB type and an arc-bonititic-type melt (Figure 13A–E). Therefore, based on the MORB melt composition, the geodynamic environment of some of the analyzed Kabaena Island chromitites fits with those proposed by Isozaki et al. [3] and Maruyama et al. [7]. According to these authors, the ultramafic rocks exposed in Kabaena Island displayed a MOR affinity and formed in a spreading zone between the Indian and Australian plates. In particular, Kabaena Island peridotites, being dominated by lherzolite, were classified as Liguria-type MORB ophiolite [4,7]. However, the composition of the chromite reported from the Italian Bracco complex, the only example of chromitite associated with the Liguria-type MORB ophiolite, is totally different from those of Kabaena Island, being Al-rich with Cr# values comprised between 0.40 and 0.55 and enriched in $TiO_2$ (up to 0.82 wt%) [47].

In this contribution, the discrepancy of the tectonic setting evinced by the mineral chemistry in one hand and by the calculated melt composition on the other hand does not allow to propose a precise model for the formation of Kabaena Island chromitites. However, Maruyama et al. [7] observed a great variation of Cr# of chromite in several peridotites of the East Sulawesi ophiolite and correlated it with different degrees of partial melting of the mantle peridotite in response to changes in the tectonic setting. More precisely, these authors reported that the majority of peridotite from Kabaena Island is consistent with a MOR origin, whereas, in some areas, the peridotites are very similar to those formed in an SSZ oceanic plateau [7]. Therefore, we can argue that the studied chromitites of Kabaena Island formed in the mantle of the forearc ophiolite located in a very dynamic area characterized by relatively quick changes in the tectonic setting varying from SSZ backarc to forearc to moderate MORB, as reported by Furnes et al. [62].

*5.2. PGM in Kabaena Island Chromitites: Their Evolution from the Magmatic Stage to Low Temperature*

The PGM found in Kabaena Island chromitites are very rare, tiny, and consist only of phases containing the most refractory PGE, i.e., Os, Ir, and Ru (Figure 10A–C and Figure 11A,B), as typical of most of the mantle hosted ophiolitic chromitites [26–31,46]. Considering their mode of occurrence, such as included in fresh chromite crystal or associated with altered minerals, and their shape, i.e., polygonal or irregular, the PGM in Kabaena Island chromitite can be classified as primary PGM that crystallized in the high-temperature magmatic stages before or during the chromite precipitation and secondary PGM that were altered and reworked during low-temperature processes. The primary PGM assemblage found in Kabaena Island chromitites is quite monotonous, consisting only of unaltered laurite (Figure 10A–C). Therefore, we can use the composition of the analyzed

laurite to define how it crystallized at high temperatures. The most important parameters that control the precipitation of magmatic Os, Ir, and Ru bearing PGM are temperature and sulfur fugacity. In particular, laurite precipitated in equilibrium with Os-Ir-(Ru) alloys at a temperature around 1300 °C and relatively low sulfur fugacity [26–28,46]. Laurite becomes progressively enriched in Os with the decreasing temperature and increasing sulfur fugacity, up to the stability field of erlichmanite. At Kabaena Island, the primary PGM assemblage is exclusively composed of laurite that is not particularly enriched in Os and characterized by a narrow variation of Os and Ru (Figure 12). Therefore, we can suggest that the sulfur fugacity was well below the Os-OsS2 buffer but was high enough to prevent crystallization of Os-Ir-(Ru) alloys during the magmatic formation of laurite and its host chromite at temperatures above 1000 °C.

The secondary PGM assemblage comprises irregular in shape and very tiny grains of Ir-Os alloy that occur associated awaruite (Figure 11A,B). They form small blebs, about 1 μm in size, located at the border of the host awaruite in contact with the silicate matrix composed of olivine and serpentine (Figure 11A,B). Serpentinization of ultramafic rocks produces fluids enriched in $H_2$ resulting from the reduction of $H_2O$, and as a consequence, peridotites have a strong reducing potential [63]. Awaruite is a typical alteration product that precipitated during the serpentinization by desulfurization in a reducing environment of magmatic Fe-Ni sulfides during the interaction between abyssal peridotite and seawater [64–66].

Alternatively, some authors suggested that awaruite was formed by the release of Ni and Fe from olivine during the serpentinization [67]. The temperature of crystallization of awaruite found associated with the chromitites of the Vourions ophiolite, Greece, was estimated to be below 445 °C [68]. The presence of small grains of primary pentlandite occurring in Kabaena Island chromitites suggests that the studied awaruite very likely formed by the desolfuration of the primary Ni-Fe sulfide. Very likely, trace amounts of Ir and Os were originally incorporated in a solid solution in the magmatic pentlandite. During serpentinization, pentlandite underwent alterations to generate awaruite plus the Ir-Os droplets, located at the contact between awaruite and silicate (Figure 11A,B). Therefore, the small blebs of Ir-Os found in Kabaena Island chromitite represent an exsolution product formed during alteration and partial desulfurization of a magmatic PGE-rich pentlandite, as previously reported for the small spot of irarsite described in the Sulawesi chromitites [38].

## 6. Conclusions

The studied small podiform chromite deposits of Kabaena Island consist of Cr-rich chromitite similar to boninite-derived chromitites formed in SSZ ophiolite complexes. The calculated composition of parental melt is consistent with the formation of Kabaena Island chromitite in a very active geodynamic area.

The composition of the associated silicates, olivine and clinopyroxene, reflects a mantle origin in a forearc ophiolite. The absence of abundant hydrous silicates suggests that a prolongated mantle metasomatism responsible for the formation of large chromite deposits was hampered.

The PGM in Kabaena Island chromitites are typical of the mantle-hosted ophiolitic chromitites. They comprise phases, such as laurite, that crystallized at magmatic temperature prior or concomitantly with the host chromite and few alloys in the Os-Ir system, formed during a low-temperature alteration process, very likely during the oceanic serpentinization that affected the study area.

**Author Contributions:** All the authors wrote and revised the paper and provided contributions to the data interpretation. The field work was carried out by S.S. and A.I., assisted by H.H. F.Z. conducted the WDS analyses. G.G. revised the data and provided the mineralogical calculation. All authors have read and agreed to the published version of the manuscript.

**Funding:** This research didn't receive any external funding.

**Data Availability Statement:** No applicable.

**Acknowledgments:** The authors acknowledge the Department of Geological Engineering, Universitas Gadjah Mada and the management of PT. Bumi Suksesindo (PT. BSI) for their supports and made possible this work completed for publication. The University Centrum for Applied Geosciences (UCAG) is thanked for the access to the Eugen Friedrich Stumpfl electron microprobe laboratory. Many thanks are due to three referees for their useful comments.

**Conflicts of Interest:** The authors declare no conflict of interest.

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
