# Peer review of "Mineralogical, Textural and Chemical Characteristics of Ophiolitic Chromitite and Platinum Group Minerals from Kabaena Island (Indonesia): Their Petrogenetic Nature and Geodynamic Setting"

_minerals, doi:10.3390/min12050516_

Round 1

Reviewer 1 Report

The work presented in this manuscript seeks to better understand the petrogenetic nature of the parental melt from which the chromitite of Kabaena Island precipitated and geodynamic setting formation of the Kabaena ophiolite and in general Sulawesi ophiolite.

I got acquainted with the works regarding East Sulawesi ophiolites of previous researchers. The East Sulawesi ophiolites have heterogeneous composition and formed at different geodynamic setting, that why the study Kabaena ophiolite (the part of Sulawesi ophiolite) and chromitite, PGM mineralization is important. The first for this region, Authors carry out systematic study the chromitites and platinum group minerals. Authors obtained data on the composition of parental melts and sulfur fugacity, which characterizes the mantle source and the geodynamic regime in a single segment of the suprasubduction zone. The authors used a modern methodological approach to the study of chromitites and PGE mineralization of the Kabaena ophiolite.

I would recommend to use accessory chromspinels from host mantle peridotites for (from  Kadarusman et al (2004) or own data) for strengthen the argumentation in discussing genetic points of the Kabaena ophiolite and chromitites. Also if it is possible to calculate oxygen fugacity.

Detailed remarks see below and

In the Abstract:

Line 16: There is an mistake in the Cr# formula: Cr# = Cr/(Cr+Al)

In the Introduction:

Line 21: You characterized the composition of olivine as mantle, and according to the composition of clinopyroxene, you are talking about geodynamic settings. You characterize minerals in different categories.

Line 27: I would just say that chromitites formed in the mantle of a the forearc region in suprasubduction setting. In this work you do not discuss the velocity change of geodynamic regime.

Line 62: Figure 1 is not Geological Map, but this is Simplified geological schem. The geological map and graphical symbols should be meet standart international requirements: color, straight-line tint, symbols. I believe that in figure 1 you need to mark geographic coordinates. It would be convenient for understanding to indicate the geographical location of the mentioned objects (for example Muna Island, Banda Peninsula)

Line 92: Reference….

Line 116: Similar to the Figure 1, the figure 2 is the Geological schem. In the legend of Figure 2, a summarily lithology composition of the complexes should be given. The symbol “Ku” reflects the Lithology composition, but the symbols “Tml, Km” are Specific name.

In Results

4.1 Chromitite texture and composition

Line 188: Are there data on the chemical composition of accessory Cr-spinels from host peridotites? It would be useful to give them to the article as well. In Kadarusman et al.(2004) article provides chemical data on accessory chrome spinels, olivines from host mantle peridotites, which can be used to compare with the chemical parameters of ore chromespinels from the chromitites.

Line 195: It is error: MgO av14.9, min 12.88, max13.84, the same with FeO.

It is strange, that chromite from lherzolite has Cr# higher and Al# lower, then in harzburgite

Line 220: Why do you think there is no correlation between Al-Cr in the some samples?

Line 230: In the Figure 6, you need to specify Cr2O3 wt%, TiO2 wt%, Al2O3 wt%

Line 259: The figure 8 B) slightly outside the page.

Line 283: Enrichment in Cr2O3 in clinopyroxene is probably assotiated with the influence of the chromite matrix.

Line 289 - Figure 9: You give a reference Rogkala et al., 2019 for composition fields of clinopyroxene. Rogkala, in turn, refers to Lian et al., 2016. A large bulk of data on the chemical composition of pyroxenes from the world ophiolite complexes was used to create the compositional fields and Lian’s article refers to these articles. I believe that you should to give references to primary reference sources, perhaps not all, but to some key articles.

Line 315: The figure 11 B) slightly outside the page

Line 319: The laurite enclosed in chromite shows a enriched in Os. The laurite in cracks shows a slight enrichment in Pd. These features are not discussed further.

Line 336: In my opinion, it would be useful to make the same calculations for accessory chrome spinels from the Kabaena peridotites. In mantle peridotites of world-wide ophiolite complexes, accessory Cr# in lherzolites have higher Al# and lower Cr# then in harzburgite and dunite. The accessory chrome spinels of Kabaena island also have the above mentioned characteristic (Kadarusman et al., 2004). It is possible that Cr# of accessory chromspinels is affected by the degree of partial melting, but ore chromspinels, by the composition of the melt with which the mantle peridotites interacted. Perhaps it should be discussed.

Author Response

Dear Referee 1,

Reply to the comments of referee 1 (marked in yellow; see file attached)

The work presented in this manuscript seeks to better understand the petrogenetic nature of the parental melt from which the chromitite of Kabaena Island precipitated and geodynamic setting formation of the Kabaena ophiolite and in general Sulawesi ophiolite.

I got acquainted with the works regarding East Sulawesi ophiolites of previous researchers. The East Sulawesi ophiolites have heterogeneous composition and formed at different geodynamic setting, that why the study Kabaena ophiolite (the part of Sulawesi ophiolite) and chromitite, PGM mineralization is important. The first for this region, Authors carry out systematic study the chromitites and platinum group minerals. Authors obtained data on the composition of parental melts and sulfur fugacity, which characterizes the mantle source and the geodynamic regime in a single segment of the suprasubduction zone. The authors used a modern methodological approach to the study of chromitites and PGE mineralization of the Kabaena ophiolite.

I would recommend to use accessory chromspinels from host mantle peridotites for (from  Kadarusman et al (2004) or own data) for strengthen the argumentation in discussing genetic points of the Kabaena ophiolite and chromitites. Also if it is possible to calculate oxygen fugacity. We have compared our data and they are very different from those reported by from  Kadarusman et al (2004) that refer to disseminated spinels in peridotite and not to  massive chromite, that is the topic of our paper.  It was not possible to obtain reliable data based on the calculation of oxygen fugacity.

Detailed remarks see below

In the Abstract:

Line 16: There is an mistake in the Cr# formula: Cr# = Cr/(Cr+Al), Done

In the Introduction:

Line 21: You characterized the composition of olivine as mantle, and according to the composition of clinopyroxene, you are talking about geodynamic settings. You characterize minerals in different categories. Yes.

Line 27: I would just say that chromitites formed in the mantle of a the forearc region in suprasubduction setting. In this work you do not discuss the velocity change of geodynamic regime. Done

Line 62: Figure 1 is not Geological Map, but this is Simplified geological schem. The geological map and graphical symbols should be meet standart international requirements: color, straight-line tint, symbols. I believe that in figure 1 you need to mark geographic coordinates. It would be convenient for understanding to indicate the geographical location of the mentioned objects (for example Muna Island, Banda Peninsula). We have changed the figure and its caption. Muna Island is added, but Banda Penninsula or Banda is far away from this map located in the eastern part of Indonesia (near Papua).

Line 92: The references are added i.e. Hamilton [9] and Kadarusman et al. [10]

Line 116: Similar to the Figure 1, the figure 2 is the Geological schem. In the legend of Figure 2, a summarily lithology composition of the complexes should be given. The symbol “Ku” reflects the Lithology composition, but the symbols “Tml, Km” are Specific name. We add a sentence in the caption to refer to the text, where all the geological information is provided.

In Results

4.1 Chromitite texture and composition

Line 188: Are there data on the chemical composition of accessory Cr-spinels from host peridotites? It would be useful to give them to the article as well. In Kadarusman et al.(2004) article provides chemical data on accessory chrome spinels, olivines from host mantle peridotites, which can be used to compare with the chemical parameters of ore chromespinels from the chromitites. We have added the data of Kadarusman et al.(2004) in figure 7C and they differ from our data.

Line 195: It is error: MgO av14.9, min 12.88, max13.84, the same with FeO. Corrected.

It is strange, that chromite from lherzolite has Cr# higher and Al# lower, then in harzburgite. Yes, we have explained it in the text, suggesting that the partial melting was not responsible.

Line 220: Why do you think there is no correlation between Al-Cr in the some samples? This is well visible in Figure 6B.

Line 230: In the Figure 6, you need to specify Cr2O3 wt%, TiO2 wt%, Al2O3 wt%. Done

Line 259: The figure 8 B) slightly outside the page. Corrected.

Line 283: Enrichment in Cr2O3 in clinopyroxene is probably assotiated with the influence of the chromite matrix. Done

 Line 289 - Figure 9: You give a reference Rogkala et al., 2019 for composition fields of clinopyroxene. Rogkala, in turn, refers to Lian et al., 2016. A large bulk of data on the chemical composition of pyroxenes from the world ophiolite complexes was used to create the compositional fields and Lian’s article refers to these articles. I believe that you should to give references to primary reference sources, perhaps not all, but to some key articles. Done.

Line 315: The figure 11 B) slightly outside the page. Corrected

Line 319: The laurite enclosed in chromite shows enriched in Os. The laurite in cracks shows a slight enrichment in Pd. These features are not discussed further. Yes because there is not a specific explanation for this, but we have observed it.

Line 336: In my opinion, it would be useful to make the same calculations for accessory chrome spinels from the Kabaena peridotites. In mantle peridotites of world-wide ophiolite complexes, accessory Cr# in lherzolites have higher Al# and lower Cr# then in harzburgite and dunite. The accessory chrome spinels of Kabaena island also have the above mentioned characteristic (Kadarusman et al., 2004). It is possible that Cr# of accessory chromspinels is affected by the degree of partial melting, but ore chromspinels, by the composition of the melt with which the mantle peridotites interacted. Perhaps it should be discussed. We have already explained that our data are different from those of (Kadarusman et al., 2004)., and we wrote: The degree of partial melting was not an important key factor during the precipitation of the studied chromitites, since they display the same composition when they are hosted in the lherzolite or in the harzburgite.

Thank you very much for your constructive comments and inputs.

Sincerely,

Arifudin Idrus & Co-authors

Reviewer 2 Report

This paper presents briefly the geology and chemical characteristics of 8 samples from the Kabaena Island, a (relatively remote ?) little known area. This study closely follows the footsteps of a previous study (Zaccarini et al, Minerals, 2016) performed on 17 samples from Sulawasi.

It is well structured, and easy to read.

While the number of samples and geological description might seem limited, I have appreciated the opportunity to know more about this region. The bibliography on the area is scarce, and the study has a pioneering-explorating aspect. For this particular reason, despite several issues, I would encourage publication of the paper with some moderate revisions.

Maybe some of the questions and suggestions below cannot be addressed (lack of information/data...). This is fine as long as the lack of data/constrain is well stated. Also, how little is known on the area could be more strongly addressed (move lines 357-361 in the introduction?)

Throughout the paper, the chromite composition is used as an exclusive tool to decipher the geodynamic setting of origin of the peridotite massif, which is the main problematic discussed in the paper. I am not at ease with this method, which seems blindly applied without consideration of other geological constrains. At least explanations should be given about why the discussion is limited to these constrains (lack of crustal section? outcropping conditions? unclear links to Sulawesi ophiolitic massifs?...).

Comments, questions and suggestions:

Sentence lines 41 to 43 "... before the Late Cretaceous." needs a reference at the end.

Line 43 erase "very"

End of paragraph line 46 may need more than refs 3-7; what are evidences behind those conclusions? What brings uncertainties?

Lines 50 to 53: are the various ophiolitic sections continuous? Is the continuity questionable? (not clear what kind of separation is briefly described line 109)

Line 59, can the "ophiolite melange" be described? (the term suggests something similar to the Franciscan complex as described by King et al, GSA Bulletin, V115, no9, p 1097-1109, 2003)

End of line 77, ref for chromitite above the Moho (e.g. Arai et al, Contrib. Min. Petrol, 2004, 147: 145-154)

line 107 "... Cretaceous to the Middle Miocene...)

Muna Island should be on the map (fig 1)

Lines 111-113: while the Kabaena Island is indeed in the same belt as the East Sulawesi ophiolite, should it be considered as the continuity of the same ophiolite, or a different massif possibly not of the same origin?

Any idea if a crustal section was present above the Kabaena peridotite (before erosion)?

Fig 3: tbrstr17 shown twice, tbrstr33 not shown. Is the #17 very close to #4 the real 17? In several figures (such as fig 7) they seem to both plot in a slightly distinct field.

Overall I found the geological background in the introduction and the geology section a bit short. Most of the geology section seems to rely on a conference abstract from the authors workgroup (ref 30) and a work by Simanduntak (ref 32), which is certainly not of easy access to candid readers. Could Siamnduntak be brought in as a co-author, and this section be more developed?

line 142: for what was it mined ?

As much as I like having some field pictures and get a sense of what are the field conditions, Figs 4A and B do not bring much to the reader (could very well look like any kind of quarry including in sedimentary rocks, anywhere else). Is no better picture available? Something that shows serpentines veins for instance?

lines 238-239 and figure 7: Could the MOR/SSZ/Forearc/etc fields in fig 7 (and others) defined by clouds of actual data presented in light grey? In particular, on what is based the forearc peridotite field? Dredged/drilled samples from several places? Ophiolite peridotite interpreted as originally from a forearc? I might be wrong but I doubt it relies on many samples, given how poorly this geodynamic setting is accessible/explored.

Cr# in MOR at the EPR and MAR that I know of, range from 0.5 to 0.6, thus close to a subset of your data. (Arai and Matsukage, Lithos 43, 1-14, 1998; Abe, J of Mine and Petrol Sci, Vol 106, 97-102, 2011)

I may have missed it in the paper: In fig 6;7 parts of the data seem to plot separately (suggesting 2 separate small groups). It should be checked that the groups do not correlate with a geographic position (N, S or W of map in fig 3) or the geological setting (lhz or hzb or contact). Whether or not that is the case, it should be said somewhere.

lines 357-361 could be in the introduction

lines 373-375 with ref 50 is a bit outdated now, most -if not all- lherzolite have been re-interpreted as refertilized mantle rather than products of low degrees melting of fertile peridotite (Leroux et al, EPSL, 259, 599-612, 2007 and later works by many others)

End of line 376 missing a reference for this statement. Actually the Mirdita ophiolite contains lherzolite, and has very important chromite mines.

Line 393 "the forearc of a SSZ ophiolite" is confusing/incorrect.

Lines 388/440 "the forearc portion of a SSZ ophiolite" is a clearer naming than line 393. However why not using the name "forearc ophiolite".

All in all, the SSZ term is confusing as it originally pointed ophiolites that had arc chemistry, without the structure of an arc, thus suggesting crustal spreading above a subducting slab that did not have an arc. Now it seems to design any ophiolite that may have originated near a subduction zone, without a proper idea of exactly where relative to the subduction trench. If a forearc setting is clearly defined, why use the term SSZ then? It is simply forearc.

Can a contemporaneous arc be associated to the ophiolite? Yes or no, this should be said somewhere.

The field of measurements exposed in the former 2016 Minerals paper (ref 41) by the same work group of authors could be shown in the figures of the submitted paper, and discussed for comparison.

I hope these comments help, and I look forward to see the paper published.

Reviewer 3

Author Response

Reply to Referee 3

We would like to thank the referee for his/her useful comments that we have used to improve our manuscript.

Here are our replies to the comments.

Comments and Suggestions for Authors

This paper presents briefly the geology and chemical characteristics of 8 samples from the Kabaena Island, a (relatively remote?) little known area. This study closely follows the footsteps of a previous study (Zaccarini et al, Minerals, 2016) performed on 17 samples from Sulawasi.

It is well structured, and easy to read.

While the number of samples and geological description might seem limited, I have appreciated the opportunity to know more about this region. The bibliography on the area is scarce, and the study has a pioneering-explorating aspect. For this particular reason, despite several issues, I would encourage publication of the paper with some moderate revisions.

Maybe some of the questions and suggestions below cannot be addressed (lack of information/data...). This is fine as long as the lack of data/constrain is well stated. Also, how little is known on the area could be more strongly addressed (move lines 357-361 in the introduction?) DONE

Throughout the paper, the chromite composition is used as an exclusive tool to decipher the geodynamic setting of origin of the peridotite massif, which is the main problematic discussed in the paper. I am not at ease with this method, which seems blindly applied without consideration of other geological constrains. At least explanations should be given about why the discussion is limited to these constrains (lack of crustal section? outcropping conditions? unclear links to Sulawesi ophiolitic massifs?...). In this contribution we report a first systematic study of chromite, silicates and PGM composition, by electron-microprobe in the podiform chromitite of Kabaena Island. . Since the origin of the Kabaena ultramafic rocks is still matter of discussion [10] and due to the complicate outcropping conditions that limited the field observation, the data are used to better understand the petrogenetic nature of the parental melt from which the chromitites of Kabaena Island precipitated and, indirectly, define the geodynamic tectonic setting of their emplacement. 

Comments, questions and suggestions:

Sentence lines 41 to 43 "... before the Late Cretaceous." needs a reference at the end. DONE

Line 43 erase "very" DONE

End of paragraph line 46 may need more than refs 3-7; what are evidences behind those conclusions? What brings uncertainties? DONE

Lines 50 to 53: are the various ophiolitic sections continuous? Is the continuity questionable? (not clear what kind of separation is briefly described line 109). Sorry but is not possible to provide a precise answer.

Line 59, can the "ophiolite melange" be described? (the term suggests something similar to the Franciscan complex as described by King et al, GSA Bulletin, V115, no9, p 1097-1109, 2003). Sorry we cannot answer to this question, because the study of the mélange of Sulawesi is far form the topic of our investigation. We have cited a paper in which other authors have mentioned this geological formation.

End of line 77, ref for chromitite above the Moho (e.g. Arai et al, Contrib. Min. Petrol, 2004, 147: 145-154) DONE

line 107 "... Cretaceous to the Middle Miocene...) DONE

Muna Island should be on the map (fig 1).DONE

Lines 111-113: while the Kabaena Island is indeed in the same belt as the East Sulawesi ophiolite, should it be considered as the continuity of the same ophiolite, or a different massif possibly not of the same origin?  Very likely is the same belt as we wrote in the text: According to Simandjuntak et al. [34], Kabaena Island is part of the East Ophiolite Sulawesi Belt, and mostly consists of ultramafic rocks, representing residual mantle.

Any idea if a crustal section was present above the Kabaena peridotite (before erosion)? Sorry no.

Fig 3: tbrstr17 shown twice, tbrstr33 not shown. Is the #17 very close to #4 the real 17? In several figures (such as fig 7) they seem to both plot in a slightly distinct field. Corrected

Overall I found the geological background in the introduction and the geology section a bit short. Most of the geology section seems to rely on a conference abstract from the authors workgroup (ref 30) and a work by Simanduntak (ref 32), which is certainly not of easy access to candid readers. Could Siamnduntak be brought in as a co-author, and this section be more developed? Has already mentioned, the data available on the studied area are limited due to difficult to carry out field trip in these remote areas and the vegetation. Nevertheless two of the authors (Arifudin Idrus and Sara Septiana) have worked a lot in the field in order to describe the geology . We have cited all the available bibliography on the geology of this poorly studied area. Sorry but we cannot do more than this.

line 142: for what was it mined? DONE

As much as I like having some field pictures and get a sense of what are the field conditions, Figs 4A and B do not bring much to the reader (could very well look like any kind of quarry including in sedimentary rocks, anywhere else). Is no better picture available? Something that shows serpentines veins for instance?  As mentioned several times, the study area is very complicate being covered by a dense vegetation. Furthermore, the outcrops of serpentinized and strongly tectonized peridotite, are always, very chaotic. We have changed two images and actually, these pictures are showing the real field conditions.

lines 238-239 and figure 7: Could the MOR/SSZ/Forearc/etc fields in fig 7 (and others) defined by clouds of actual data presented in light grey? In particular, on what is based the forearc peridotite field? Dredged/drilled samples from several places? Ophiolite peridotite interpreted as originally from a forearc? I might be wrong but I doubt it relies on many samples, given how poorly this geodynamic setting is accessible/explored. These data are taken from the literature and we used them. We have cited the original sources. Even if they were based on few samples,  they are available in the international literature and published in international journals. This is the reason why we used them

Cr# in MOR at the EPR and MAR that I know of, range from 0.5 to 0.6, thus close to a subset of your data. (Arai and Matsukage, Lithos 43, 1-14, 1998; Abe, J of Mine and Petrol Sci, Vol 106, 97-102, 2011). Sorry we did not understand this comments.

I may have missed it in the paper: In fig 6;7 parts of the data seem to plot separately (suggesting 2 separate small groups). It should be checked that the groups do not correlate with a geographic position (N, S or W of map in fig 3) or the geological setting (lhz or hzb or contact). Whether or not that is the case, it should be said somewhere.  We have explained it in the text.

lines 357-361 could be in the introduction DONE

lines 373-375 with ref 50 is a bit outdated now, most -if not all- lherzolite have been re-interpreted as refertilized mantle rather than products of low degrees melting of fertile peridotite (Leroux et al, EPSL, 259, 599-612, 2007 and later works by many others). Sorry we gave credit to old researchers especially when they were among the first to propone a model and, more important, in the specific case, we cited articles that are reporting on  the petrographic aspects of ophiolite and their link with the formation of chromitite that is the topic of our paper.

End of line 376 missing a reference for this statement. Actually the Mirdita ophiolite contains lherzolite, and has very important chromite mines. We wrote Large chromite deposits have been preferentially described in the mantle sections of the HOT ophiolite. Preferentially and not exclusively.

Line 393 "the forearc of a SSZ ophiolite" is confusing/incorrect. Sorry, we also do not fully understand this comment. This is what is simply reported in several geological books  ….. During subduction, an oceanic plate is thrust below another tectonic plate, which may be oceanic or continental. Water and other volatiles in the down-going plate cause flux melting in the upper mantle, creating magma that rises and penetrates the overriding plate, forming a volcanic arc. The weight of the down-going slab flexes the down-going plate creating an oceanic trench. The area between the trench and the arc is the forearc region, and the area behind the arc (i.e. on the side away from the trench) is the back-arc region. ……. Nevertheless, following the suggestion of the referee, we have simplified the term forearc of a SSZ ophiolite with forearc ophiolite.

Lines 388/440 "the forearc portion of a SSZ ophiolite" is a clearer naming than line 393. However why not using the name "forearc ophiolite". DONE

All in all, the SSZ term is confusing as it originally pointed ophiolites that had arc chemistry, without the structure of an arc, thus suggesting crustal spreading above a subducting slab that did not have an arc. Now it seems to design any ophiolite that may have originated near a subduction zone, without a proper idea of exactly where relative to the subduction trench. If a forearc setting is clearly defined, why use the term SSZ then? It is simply forearc. See comment above

Can a contemporaneous arc be associated to the ophiolite? Yes or no, this should be said somewhere. Sorry we do not understand this comment and we cannot answer.

The field of measurements exposed in the former 2016 Minerals paper (ref 41) by the same work group of authors could be shown in the figures of the submitted paper, and discussed for comparison. DONE

I hope these comments help, and I look forward to see the paper published.

Thank you very much for your constructive comments and suggestion.

Reviewer 3

Authors

Reviewer 3 Report

Journal: Minerals (ISSN 2075-163X)

Manuscript ID: minerals-1678891

Type: Article

Title: Mineralogical, Textural and Chemical Characteristics of Ophiolitic Chromitite and Platinum Group Minerals from Kabaena Island (Indonesia): Their Petrogenetic Nature and Geodynamic Setting

Authors: Arifudin Idrus, Sara Septiana, Federica Zaccarini, Giorgio Garuti, Hasria Hasria

Section: Mineral Geochemistry and Geochronology

Special Issue: Platinum-Group Elements and Platinum-Group Minerals: Inferences from Field, Geochemical, Mineralogical, and Petrological Findings

General comments for authors:

The work presented by the authors is interesting and could be accepted by readers. However, there are some small remarks that the authors have to take into account.

Specific comments for authors:

Abstract: The Abstract needs to be rewritten. The form in which it appears is not well structured. The authors have to start with a short introduction; then clearly state the objectives to be developed with this work; next, the methods applied to meet these objectives should be mentioned; following this, the results obtained should be briefly mentioned; finally, an idea should be given on how these results could be applied. Please rewrite.

Line 21. Write the full phrase "supra-subduction zone" and then the abbreviation "SSZ"; this is done in the opening paragraphs. Subsequently, the abbreviation will suffice. Check.

Figure 1. Authors have to indicate the study area on the map. Check.

Lines 118 to 152. These arguments should be transferred to a new Section entitled: "Materials". Check.

Section 3. Authors are encouraged to change the phrase "Research Methods" to "Methods". Please check.

Figure 5. Authors are encouraged to write the meaning of abbreviations at the bottom of the figure. Please check.

Figure 10. Authors are encouraged to write the meaning of abbreviations at the bottom of the figure. Please check.

Section 6. Authors are encouraged to change the phrase "Concluding Remarks" to "Conclusions". Please check.

Author Response

Dear Referee 2,

Reply to the comments of referee 2 (see attached file)

General comments for authors:

The work presented by the authors is interesting and could be accepted by readers. However, there are some small remarks that the authors have to take into account.

Specific comments for authors:

Abstract: The Abstract needs to be rewritten. The form in which it appears is not well structured. The authors have to start with a short introduction; then clearly state the objectives to be developed with this work; next, the methods applied to meet these objectives should be mentioned; following this, the results obtained should be briefly mentioned; finally, an idea should be given on how these results could be applied. Please rewrite. Done.

Line 21. Write the full phrase "supra-subduction zone" and then the abbreviation "SSZ"; this is done in the opening paragraphs. Subsequently, the abbreviation will suffice. Check. Done.

Figure 1. Authors have to indicate the study area on the map. Check. We have clearly indicated the position of Kabaena Island.

Lines 118 to 152. These arguments should be transferred to a new Section entitled: "Materials". Check. Done.  

Section 3. Authors are encouraged to change the phrase "Research Methods" to "Methods". Please check. Done.

Figure 5. Authors are encouraged to write the meaning of abbreviations at the bottom of the figure. Please check. Done.

Figure 10. Authors are encouraged to write the meaning of abbreviations at the bottom of the figure. Please check. Done.

Section 6. Authors are encouraged to change the phrase "Concluding Remarks" to "Conclusions". Please check. Done.

Thank you very much for your inputs and suggestion.

Sincerely,

Arifudin Idrus and Co-authors

NOTE: The attached revised manuscript (highlighted by yellow color) is also directed for Referee 1. Thank you.
